# Impact of the eruption of Mt. Pinatubo on the chemical composition of the stratosphere

Markus Kilian[1], Sabine Brinkop[1], and Patrick Jöckel[1]

[1]Deutsches Zentrum für Luft- und Raumfahrt (DLR), Institut für Physik der Atmosphäre, Oberpfaffenhofen, Germany

**Correspondence:** Markus Kilian (markus.kilian@dlr.de)

**Abstract.** This article describes the volcanic effect of the Mt. Pinatubo eruption in June 1991 on the ozone ($O_3$) and methane ($CH_4$) distribution in the stratosphere, as simulated with the chemistry-climate model EMAC (ECHAM/MESSy Atmospheric Chemistry, ECHAM5, version 5.3.02, MESSy, version 2.51). In this study, the effects of volcanic heating and heterogeneous chemistry on the chemical composition, caused by the volcanic aerosol, are separated. Global model simulations over the relevant period of the eruption from 1989 to 1997 with EMAC in T42L90MA resolution with specified dynamics and interactive chemistry were performed. The first simulation (VOL) contains the volcanic perturbation as additional aerosol load and thus simulates the interaction of the aerosol with the chemistry and the radiation. The second simulation (NOVOL) neglects the eruption and represents the undistorbed atmosphere. In the third simulation (CVOL) the volcanic aerosol only interacts with the heterogeneous chemistry, such that volcanic heating is omitted. The differences between the simulation results VOL-NOVOL describe the total effect of the Mt. Pinatubo eruption on the chemical composition, VOL-CVOL the stratospheric heating effect and CVOL-NOVOL the chemical effect of the aerosol on the heterogeneous chemistry. The post volcanic stratosphere shows a decrease of the $O_3$ column in the tropics, and an increase in the mid-latitudes and polar regions, lasting roughly one year. This change in the ozone column is solely a result of the heating effect. The subsequent decrease of the ozone column is related to the chemical effect. The contribution of the catalytic loss cycles ($NO_x$, $HO_x$, $ClO_x$ and $BrO_x$) on the depletion of $O_3$ is analysed in detail. In the tropics, $CH_4$ increases in the upper stratosphere because of stronger vertical transport.

## 1 Introduction

Major volcanic eruptions in the tropics are able to inject material into the stratosphere and perturb the physical and the chemical states of the climate system for several years and longer (Robock, 2000; Thomas et al., 2009; Muthers et al., 2015). The emissions are mainly ash and sulphur in the form of $SO_2$, which oxidizes to sulphate in the stratosphere, and sulphate aerosols are formed within weeks. The Mt. Pinatubo eruption in June 1991 decreased the tropical ozone column significantly, by 13-20 DU (Grant et al., 1992). Stratospheric ozone is a well studied greenhouse gas with a great public interest because it absorbs UV radiation and protects human health from harmful radiation. The abundance of ozone is mainly controlled by the production via photolysis and depletion by catalysts such as $NO_x$, $ClO_x$, $BrO_x$ and $HO_x$ (Solomon, 1999). The yearlong ozone production in the tropics and the transport via the Brewer-Dobson circulation (BDC) towards the polar regions, together with the ozone

depletion due to the temperature dependent PSC's (polar stratospheric clouds) and the resulting activation of chlorine explain the global distribution of $O_3$.

The eruption of the Mt. Pinatubo on the 12th of June 1991 was one of the strongest volcanic eruptions in the 20th century (Thomas et al., 2009). It affected the chemical composition of the atmosphere considerably (Textor et al., 2004; von Glasow et al., 2009; Poberaj et al., 2011; Aquila et al., 2013). The formed sulphate aerosols were rapidly zonally dispersed and transported into both hemispheres by the BDC. They absorb the solar and the terrestrial infrared radiation, which results in a heating of the stratosphere and a cooling of the troposphere beneath (Labitzke and McCormick, 1992). Stenchikov et al. (1998) established that the near-infrared solar forcing mainly drives the stratospheric heating due to an increase in the atmospheric extinction by volcanic aerosol. Brühl et al. (2015) performed EMAC simulations for the Pinatubo eruption to show the importance of radiative feedback on the dynamics. They stated, that radiative heating by volcanic aerosols enhances tropical upwelling and accelerates the tape-recorder. After the Mt. Pinatubo eruption, the temperature in the lower and middle stratosphere increased by up to 3.5 K at 30 hPa and decreased by 0.5 K in the troposphere (Labitzke and McCormick, 1992; Self et al., 1999; Robock, 2000; von Glasow et al., 2009). The volcanic heating by the Mt. Pinatubo eruption lead to an increase of the vertical ascent by 20 % and subsequent poleward transport of air from regions with low ozone and thus reduced the ozone column in the tropics (McCormick et al., 1995b; Dameris et al., 2005). Moreover, volcanic heating modifies the ozone budget via an acceleration of the temperature dependent reaction rates. Ozone measurements in the post-volcanic atmosphere of the Mt. Pinatubo eruption showed a total ozone column decrease by 6-8 % in the tropics within a month after the eruption (Grant et al., 1992; Schoeberl et al., 1993a; McCormick et al., 1995b). McCormick et al. (1995b) figured out that the bulk of the ozone loss after the Mt. Pinatubo eruption appeared between 24-25 km with up to 20 %. The larger liquid aerosol surface accelerates the heterogeneous reactions and affects the $O_3$ chemistry, e.g. converting $NO_x$ into $HNO_3$, provides and activates chlorine from reservoirs, and alters catalytic ozone destruction (Solomon et al., 1996; Solomon, 1999; Robock, 2000; Poberaj et al., 2011; Aquila et al., 2013).

Modelling studies on the eruption of Mt. Pinatubo aim to understand which processes are responsible for the ozone changes and how these changes affect the dynamics. These studies differ with respect to the methodology. Rozanov et al. (2002) simulated the volcanic induced ozone depletion using a GCM including interactive chemistry and stratosphere-troposphere interactions. Two ensembles of free-running model simulations were performed. One ensemble simulates the volcanic aerosol, hence including both, the chemical and dynamical effects, and the second set uses no volcanic forcing. Likewise, a set of free-running model simulations, but with a coupled atmosphere-ocean-chemistry model has been used by Muthers et al. (2015). They were also interested in the role of different climate settings (specific atmospheric composition of greenhouse gases) and performed corresponding sensitivity studies.

Aquila et al. (2013) used ensembles of 10 free-running model simulations representing either the case of full interaction of volcanic aerosol with chemistry and radiation, with chemistry only, with radiation only and one simulation without the eruption effect. They could attribute the change in ozone to dynamical and chemical effects in the post Pinatubo period. Telford et al. (2009a) combined two nudged simulations and one free-running simulation in order to distinguish between chemical and dynamical effects on ozone of the Mt. Pinatubo eruption. Their first simulation nudged towards ERA-40 uses all aspects of

the model with prescribed volcanic aerosol. The second nudged simulation is performed without the Pinatubo, but with an unperturbed background aerosol. The difference between both lead to the so-called chemical effect on ozone. Admittedly, the chemical effect of Telford et al. (2009b) is not purely chemical, because it still consists of both, a volcanic heating with subsequent influence on the stratospheric dynamics, and a perturbation of the heterogeneous chemistry. In our present study we separate Telford et al's alleged chemical effect into the heating through absorption by the volcanic aerosol and the pure chemical effect, the latter just developing from a change of the heterogeneous chemistry by the aerosol surface. In our study Telford et al's chemical effect is called combined effect. The separation of both allows a better understanding and quantification of the detailed chemical processes concerning ozone. The new findings can be used to optimize the representation of volcanic eruptions in chemistry climate models. Muthers et al. (2015) also simulated the temperature driven effects decoupled from the chemical effect with a coupled atmosphere–ocean–chemistry–climate model, but did not separate the chemical effect of the heterogeneous chemistry from the heating effect of the volcanic heating.

Here, we performed global simulations of the eruption from 1989 to 1997 with specified dynamics and full chemistry with the chemistry-climate model EMAC. One simulation contains the volcanic perturbation as an additional aerosol load (VOL) and thus the full interaction of the aerosol with the chemistry and the radiation. In the second simulation, only the chemistry on the volcanic aerosols is accounted for (CVOL), and the third simulation neglects the eruption entirely and represents the undisturbed atmosphere (NOVOL). The differences of the simulation results (VOL-NOVOL) describe the combined effect of the Mt. Pinatubo eruption, (VOL-CVOL) of stratospheric heating effect only, and (CVOL-NOVOL) solely the effect of the chemistry due to the additional aerosol chemical effect. Moreover, the effect on the relevant chemical cycles is calculated between the CVOL and NOVOL simulations.

Section 2 provides a description of the chemistry-climate model EMAC, the setup of the simulations and the methodology of the analysis. Section 3 evaluates the EMAC results with the SWOOSH satellite measurements data set. Section 4 describes all relevant physical and chemical parameters, which are perturbed by the volcanic aerosol. Section 5 discusses the findings and provides an outlook on further studies. Section 6 summarises the most important results.

This is a follow-up study of Löffler et al. (2016), who analysed the perturbation of the stratospheric water vapour by the Mt. Pinatubo eruption. The motivation of this study is to differentiate the effect of the eruption on the chemical composition into:

1. the effect of stratospheric heating and subsequent changes in transport,

2. the pure chemical effect due to heterogeneous reactions on the volcanic aerosol and the modification of the relevant chemical cycles.

## 2 Model Simulations

### 2.1 Model Description

The European Centre for Medium-Range Weather Forecasts Hamburg – Modular Earth Submodel System (ECHAM/MESSy) Atmospheric Chemistry model (EMAC) is a numerical chemistry and climate simulation system that includes sub-models

describing tropospheric and middle atmosphere processes and their interaction with oceans, land, and human influences (Jöckel et al., 2010, 2016). It uses the second version of the MESSy to link multi-institutional computer codes. The core atmospheric model is the 5th generation European Centre Hamburg general circulation model ECHAM (Roeckner et al., 2006). For the present study we applied EMAC (ECHAM5 version 5.3.02, MESSy version 2.51) in the T42L90MA resolution, i.e. with a spherical truncation of T42 (corresponding to a quadratic Gaussian grid of ca. 2.8° × 2.8° in latitude and longitude) with 90 vertical hybrid pressure levels up to 0.01 hPa.

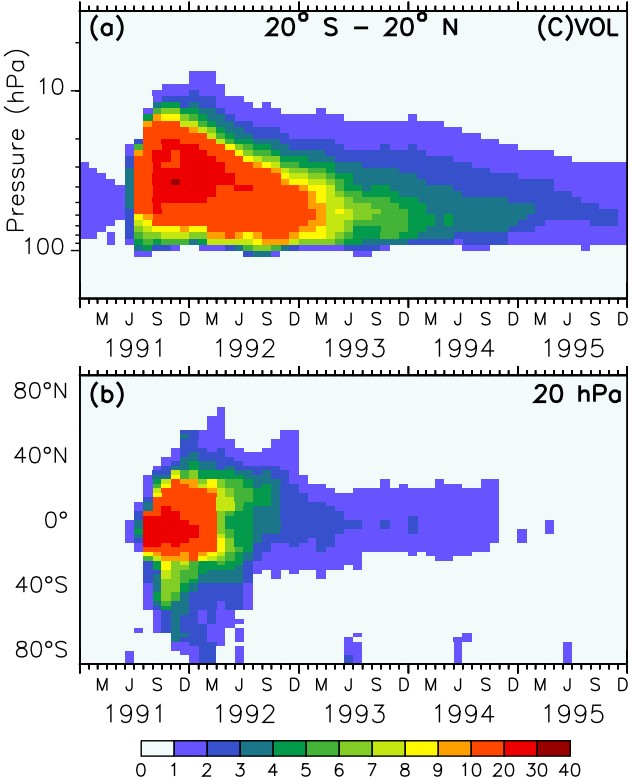

**Figure 1.** Colours show the monthly averaged liquid aerosol surface area density ($10^{-8}$ cm$^2$cm$^{-3}$) of the volcanic sulfate aerosols in simulations VOL and CVOL in the stratosphere as tropical averaged vertical time series between 20° S and 20° N **(a)**, and as latitudinal time series at 20hPa **(b)**. The intervals of the colour scale are not equidistant.

The forcing of the volcano in the model is represented by the prescribed surface area density in time and space (Figures 1 and 2). Hence, volcanic aerosols in the simulations VOL and CVOL are not treated by an interactive aerosol module, but the volcanic aerosols are based on measurements from satellites. This means, that the aerosol interacts with clouds and radiation, but does not consider the impact of the atmospheric dynamics on the aerosol distribution. The aerosol data originate from the CCMI data set, which is based on different satellite measurements (Diallo et al., 2017; Revell et al., 2017): Stratospheric Aerosol Measurement (SAM), Stratospheric Aerosol and Gas Experiment (SAGE I and II), Cloud Aerosol Lidar and Infrared Pathfinder Satellite Observations (CALIPSO), and Optical Spectrograph and InfraRed Imaging System (OSIRIS). It describes

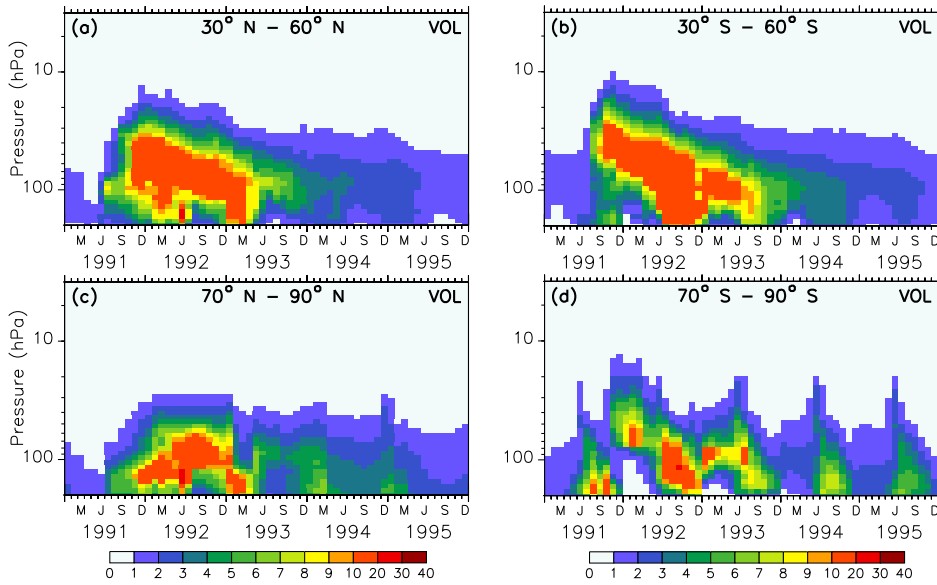

**Figure 2.** Colours show the monthly averaged liquid aerosol surface area density ($10^{-8}$ cm$^2$cm$^{-3}$) of the volcanic sulfate aerosols used as boundary conditions in simulations VOL and CVOL in the stratosphere as averaged vertical time series in the mid-latitudes (**(a)** and **(b)**) and in the polar regions (**(c)** and **(d)**). The intervals of the colour scale are not equidistant.

a single-mode log-normal aerosol size distribution derived by means of the SAGE-4$\lambda$ algorithm, which is compiled out of four wavelengths (385, 452, 525, and 1024 nm) of SAGE II data when available (Revell et al., 2017).

### 2.2 ESCiMo Consortial Simulations

The volcanic simulations of this study were performed like multiple simulations with different boundary conditions within the
5 Earth System Chemistry integrated Modelling (ESCiMo) project (Jöckel et al., 2016). These model simulations were defined to improve the understanding of processes in the atmosphere and also to help answer questions related to climate change, ozone depletion and air quality, which is an important contribution to the WMO/UNEP (World Meteorological Organization/United Nations Environment Programme) ozone and IPCC (Intergovernmental Panel on Climate Change) climate assessments (WMO, 2019).

Here we employ these simulations and assess the impacts of the Mt. Pinatubo eruption on the atmospheric chemistry. In our study we focus on 3 simulations with specified dynamics. They are branched off from a specified dynamics hindcast simulation and are "nudged" with a Newtonian relaxation technique towards 6-hourly ERA-Interim reanalysis data from the ECMWF (European Centre for Medium-Range Weather Forecasts), which is available since 1979 (Dee et al., 2011). The Newtonian relaxation (nudging) is applied to the prognostic variables divergence, vorticity, temperature, and the (logarithm of the) surface
pressure in spectral space with relaxation times $\tau_x$ of 48, 6, 24, and 24 h, respectively. The global-mean temperature ($\overline{T}$) is not affected by the nudging technique, because we exclude nudging of the "wave-0" in spectral space. Moreover, the nudging is

not applied uniformly in the vertical: the boundary layer and the stratosphere above 10 hPa are not nudged. The simulations VOL, CVOL and NOVOL span the years 1989 to 1997.

The set up (Table 1) is comparable with those of the EMAC simulations from Löffler et al. (2016), who simulated the effect of volcanic heating on SWV. In our study, VOL considers the volcanic aerosols and its interaction with both, the radiation and the chemistry, and shows the combined effect of the volcanic eruption. NOVOL omits the volcanic eruption by using an as far as possible unperturbed annual background aerosol from the year 2011. Years without any volcanic eruptions are rare, but our used background aerosol is only marginally affected by the mid-size volcanic eruption of Nabro in 2011, although sulphate aerosols of the eruption of Nabro reached altitudes of up to 18 km, measured by the Michelson Interferometer for Passive Atmospheric Sounding (MIPAS, Griessbach et al., 2016). CVOL considers the volcanic aerosols like VOL, but only the heterogeneous reactions on the aerosol surface are affected, whereas the radiative transmission remains unaffected by the volcanic aerosols.

| Simulation | Volcanic Aerosol | Aerosol Interaction |
|:---:|:---:|:---:|
| VOL | x | Radiation, Chemistry |
| CVOL | x | Chemistry |
| NOVOL | - | - |

**Table 1.** Overview of the three simulations with their differences in model set-up in terms of the volcanic perturbation and its interactions.

## 2.3 Methodology

The results are presented as differences of monthly mean values between the simulations, with exceptions being denoted clearly. The goal is to separate the heating effect of volcanic aerosols from the chemical effect caused by the altered heterogeneous chemistry on the ozone distribution. The difference VOL-NOVOL shows the combined effect of the stratospheric temperature increase due to absorption by volcanic aerosols and the intensification of the heterogeneous reactions, due to an increase of the aerosol surface density. VOL-CVOL yields the stratospheric heating effect, which includes the absorption of solar radiation by the volcanic aerosol and changes in transport, as well as the secondary change of the temperature dependent heterogeneous reaction rates. The pure chemical effect induced by the larger aerosol surface can be separated by the difference of CVOL-NOVOL. Due to the applied nudging, the temporal evolution of the synoptic conditions is similar in all 3 simulations. Using nudged simulations has several advantages over free-running simulations to study the impact on the chemistry: First, the temperature response is closer to observations, which is important, as ozone chemistry is temperature dependent and second the results appear less noisy. Our (nudged) simulation pair (VOL and NOVOL) is similar with respect to the synoptic situation, so the effect of aerosol heating on subgrid-scale chemistry and transport of ozone can be revealed more clearly. This would be more difficult, if one allows the synoptic situation to evolve freely. Moreover, for such a concept a large set of ensemble simulations is necessary (Aquila et al., 2013). In order to show how large the stratospheric temperatures in VOL and CVOL are perturbed by nudging, Löffler et al. (2016), (their supplement, Figure S2) performed two quasi free running simulations

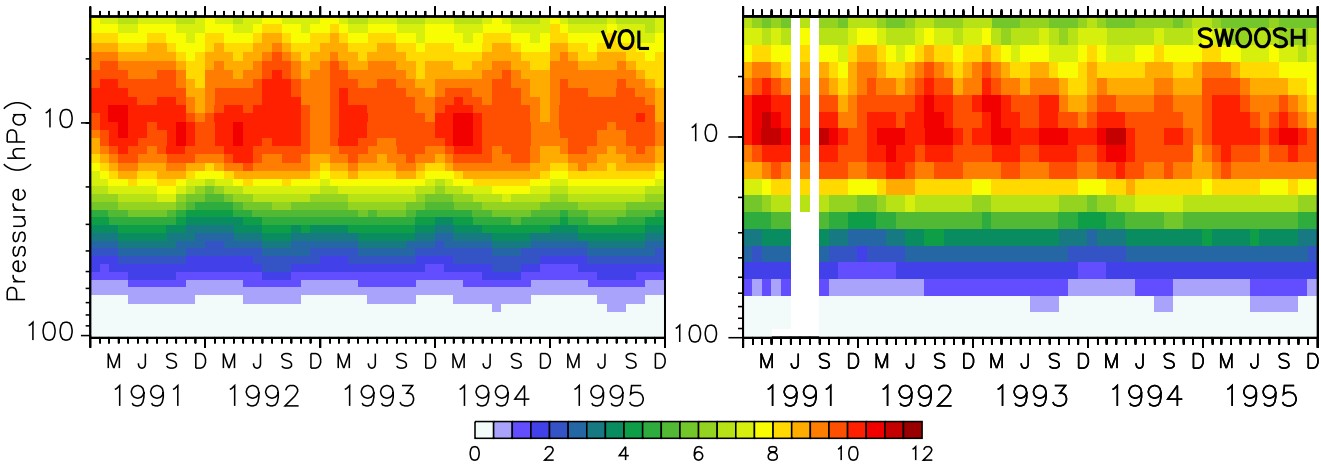

**Figure 3.** Colours show the absolute zonally and monthly averaged ozone mixing ratios (ppmv) at 10° N of the volcanic simulation VOL (left) and the SWOOSH satellite data set (right) for the period from 1991-1995 (Davis et al., 2016).

(named QF), one with volcanic aerosol heating and one without. In the QF simulation pair only the (logarithm) of the surface pressure is nudged (and SST (Sea Surface Temperature)/SIC (Sea Ice Concentration) are prescribed) to study the effect of omitting the nudging of temperature, divergence and vorticity on the results. The temperature difference of the QF simulation pair is larger compared to that of the nudged pair (about 50%, Figs. S1 and S2 of Löffler et al., 2016) and, moreover appears

noisy. As already stated by Löffler et al. (2016), nudging (of temperature, divergence and vorticity) reduces the cold point temperature during the eruption. However, the development of temperature is similar to the nudged simulation pairs. The one-by-one comparison shows the volcanic perturbations and only a sub-synoptic noise, because the effects of nudging cancel out. The production and loss rates of ozone are calculated off-line with the diagnostic tool *STratO3Bud* as described by Meul et al. (2014) and Grewe et al. (2017).

**3  Comparison with Observations**

The EMAC model has been evaluated intensively by Jöckel et al. (2006); Jöckel et al. (2010, 2016), and with a focus on the Arctic polar stratosphere by Khosrawi et al. (2017, 2018). In order to show the reliability of the simulated ozone distribution here, the volcanic simulation VOL is compared with the Stratospheric Water and OzOne Satellite Homogenized (SWOOSH) data set provided by NOAA (National Centers for Environmental Information, Davis et al., 2016), and the TOMS (Total

15 Ozone Mapping Spectrometer) data set processed by the Goddard's Ozone Processing Team arising from satellite observations (TOMS-Science-Team, 2004; Wellemeyer et al., 2004). The SWOOSH data set was constructed to investigate the variability and change of water vapour and ozone in the stratosphere. SWOOSH is a database of global long-term satellite $O_3$ and stratospheric water vapour (SWV) measurements presented on a vertical grid as monthly averages on pressure levels. The measurements from different satellite instruments, namely SAGE II, SAGE III, the Halogen Occultation Experiment (HALOE),

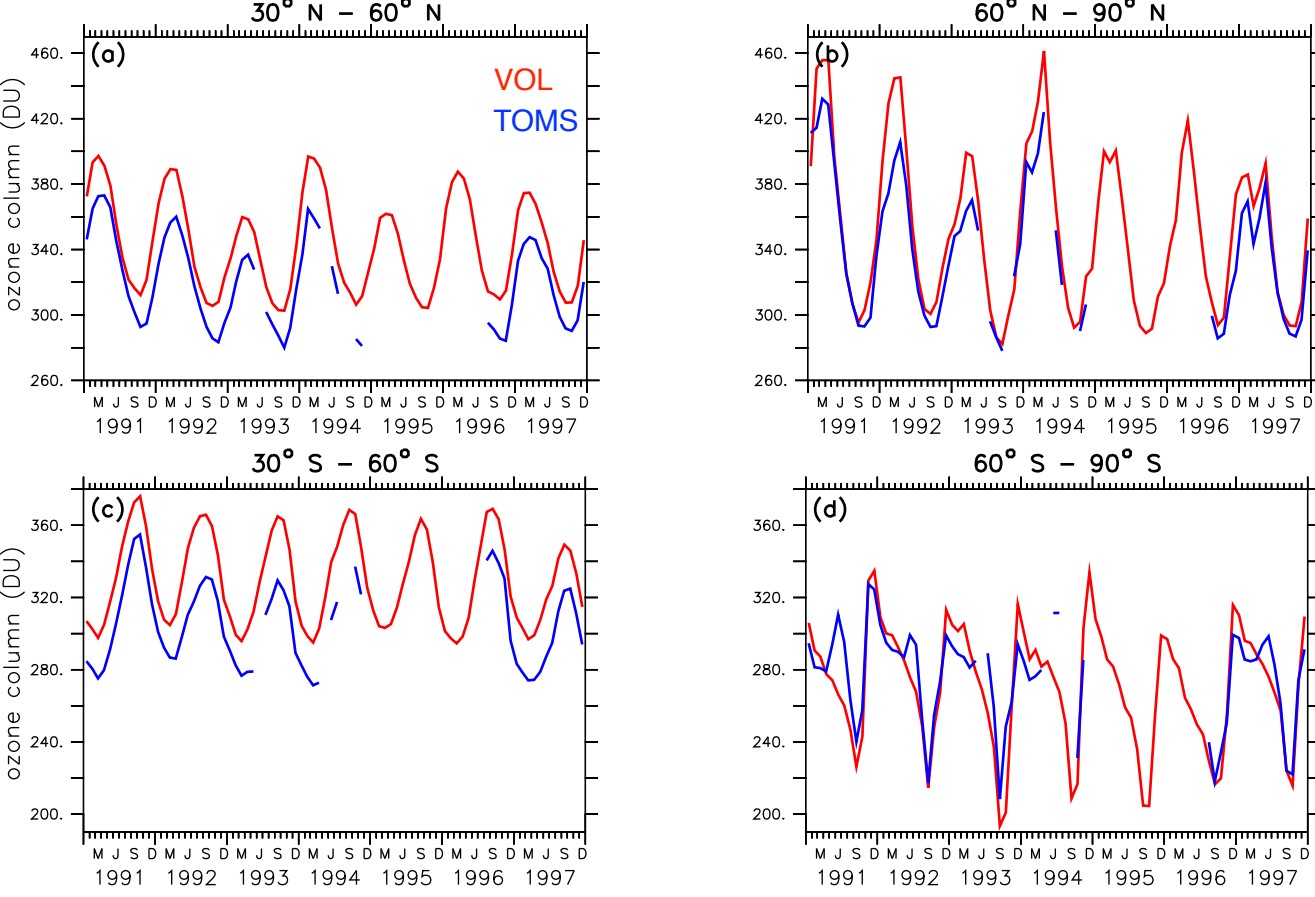

**Figure 4.** Lines show the zonally and monthly averaged total ozone column (DU) between 30° N - 60° N **(a)** and 60° N - 90° N **(b)** as well as between 30° S - 60° S **(c)** and 60° S - 90° S **(d)** of the volcanic simulation VOL, and the TOMS data for the period from 1991-1997 (TOMS-Science-Team, 2004). The period between 1994-1996 shows data gaps because the atmosphere was too opaque for reliable ozone measurements.

the Upper Atmosphere Research Satellite (UARS), the Microwave Limb Sounder (MLS), and the Earth Observing System (EOS) Aura MLS, as well as from other merged data products have been combined. The TOMS data set was constructed to investigate the distribution of the total ozone column under extreme conditions like during a strong aerosol load (Wellemeyer et al., 2004). The spatial and temporal distribution of ozone in VOL corresponds to the SWOOSH data (Figure 3). The ozone layer between 5 and 20 hPa with mixing ratios of 10-12 ppmv, as well as the seasonal cycle, are well represented. With the Mt. Pinatubo eruption in June 1991, the atmosphere was too opaque for reliable satellite measurements, and consequently, those months are marked with white stripes showing the data gap. Figure 4 shows the comparison of the total ozone column between simulations VOL and the TOMS data. The annual maxima of the total ozone column in VOL in the northern and southern midlatitudes (Figure 4a,b) is systematically overestimated by more than 20 DU in comparison to the TOMS observations. This

is consistent with the evaluation of the total ozone column of the nudged ESCiMo simulations with the BSTCO (Bodeker Scientific combined total column ozone database) observations by Jöckel et al. (2016), who also found an overestimation of total ozone in the northern and southern midlatitudes of up to 10-20 DU (Jöckel et al., 2016, Figure 27h). During the period of transition to the ozone minimum in the southern spring the difference decreases down to 5 DU. At the Antarctic polar cap the total ozone column simulated in VOL compares well with the TOMS data (Figure 4d). The ozone maxima at the North Pole during spring are stronger overestimated (up to 40 DU) by the VOL simulation than at the South Pole (10-15 DU, Figure 4b,d). The seasonal ozone cycle in VOL corresponds well with the TOMS data on both hemispheres.

## 4   Results

We show in the results section the effect of the volcanic perturbation on temperature, on the vertical ozone column, on the vertical distribution of ozone, the chemical cycles and the Polar stratospheric clouds (PSC). First, we investigate the temperature perturbation, because sulphate aerosols in the stratosphere increase the atmospheric extinction, resulting in a higher absorption of terrestrial (longwave) and solar (shortwave) radiation warming the stratosphere. Heterogeneous ozone chemistry is affected by the sulphate aerosols, which alter the ozone depletion cycles. Temperature changes at the tropopause perturb the cold point and alter the transport of SWV. An increase of the vertical transport also modifies methane. Since the ozone chemistry in the polar regions is strongly affected by the formation of Polar Stratospheric Clouds (PSC), their behavior in a post volcanic stratosphere is analysed.

### 4.1   Temperature

The strongest heating due to absorption of solar and terrestrial infrared radiation by volcanic aerosols and by the increase of ozone due to transport occurs in the middle stratosphere of the tropics (Figure 5b). Figure 5c shows the indirect volcanic heating by the perturbed heterogeneous chemistry, which changes the chemical composition like ozone and causes additional absorption. The combined effect (Figure 5a), which is defined as the sum of the transport and the chemical effect shows the strongest heating in the lower and middle stratosphere in December 1991. The vertical extent is congruent with the volcanic aerosol plume. The largest temperature increase occurs between 40-60 hPa in the tropical stratosphere. In the central panel (VOL-CVOL) the strongest heating occurs at the same height, but stronger with a maximum between 40° N and 40° S. The bottom panel displays a heating of up to 0.4 K in the upper stratosphere between 5 and 20 hPa and a cooling of up to 0.2 K between 30 and 70 hPa. More stratospheric ozone increases temperature due to more absorption of solar radiation and less ozone cools the stratosphere. Thus, the main temperature change of this volcanic eruption arises mostly from radiative absorption by the volcanic aerosol superimposed by a cooling due to reduced ozone.

### 4.2   Total Ozone Column

Our simulations show that the total ozone column decreases in the first 12 months after the eruption of the Mt. Pinatubo by up to 18 DU (6 %) in the tropics (Figure 6a). In contrast, the mid latitudes and the polar regions experience an increase of

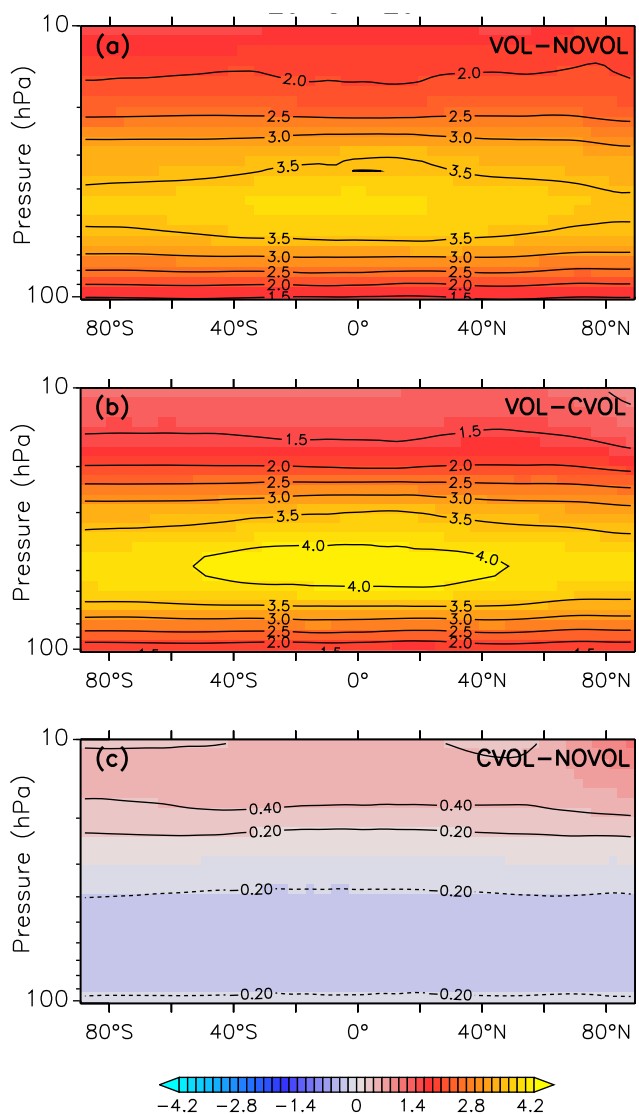

**Figure 5.** Monthly mean of the zonally averaged stratospheric temperature difference (K) between the model simulations (CVOL and NOVOL) for December 1991. **(a)** shows the combined effect, **(b)** the heating effect and **(c)** the chemical effect. Black contours indicate the isothermal curves (K). Contour intervals are 0.5 K in **(a)** and **(b)**, and 0.2 K in **(c)**.

the total ozone column by 8-12 DU (2 %), smaller in the southern than in the northern mid latitudes. In May 1992, an ozone reduction propagates towards the North Pole and in August 1992 a similar anomaly propagates towards the South Pole. The combined effect **(a)** is the sum (VOL-CVOL)+(CVOL-NOVOL)=VOL-NOVOL of the heating **(b)** and chemical effect **(c)**, thus the change of the total ozone column is caused by a superposition of both effects (**b** and **c**). The decrease of ozone in the tropics and the increase in the mid and higher latitudes in the first 2 years after the Mt. Pinatubo eruption are caused solely by the

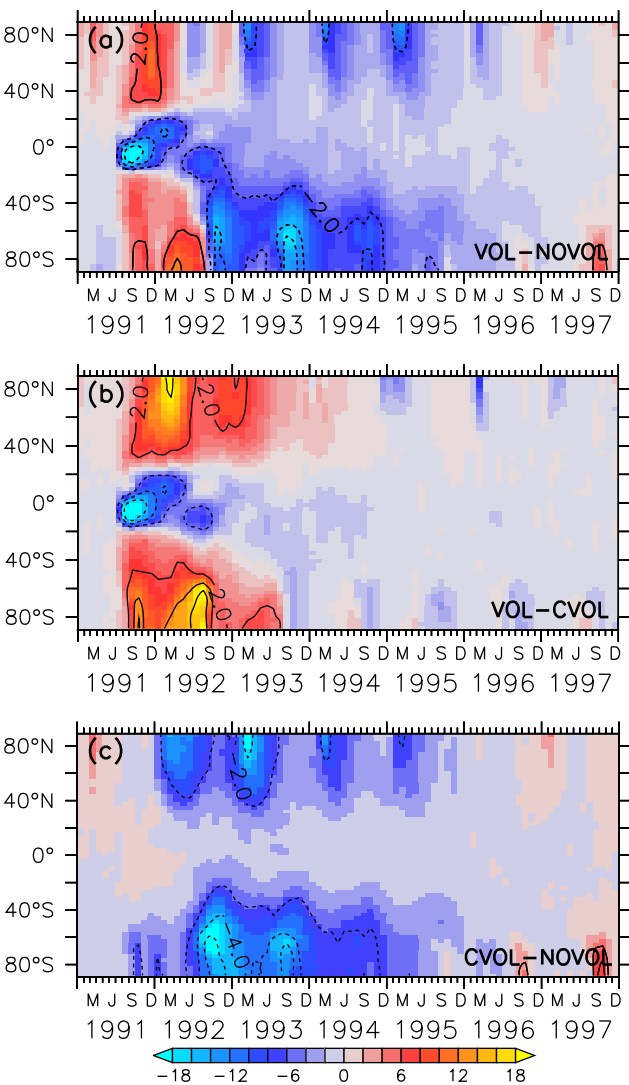

**Figure 6.** Colours show the differences of the zonal mean total ozone column (DU) as combined effect (VOL-NOVOL) **(a)**, heating effect (VOL-CVOL) **(b)**, and chemical effect (CVOL-NOVOL) **(c)** as a function of the latitude in degrees (°). Contours indicate the corresponding relative changes (%). Contour intervals are 2 %.

volcanic heating (Figure 6b), i.e. the increase in the BDC through the volcanic aerosol induced heating of the stratosphere and a subsequent increase in transport of ozone. This result is in contradiction to Poberaj et al. (2011), who attributed the absence of ozone depletion in the southern hemisphere to interannual dynamic variability.

    The ozone decrease by up to 4 % in the North Pole region and by up to 6 % in the South Pole region, starting in the respective
5   spring seasons of 1992, is due to the chemical effect alone (Figure 6c).

Thus, the heating and chemical effect alter the ozone distribution in the mid-latitudes and in the polar regions in opposite directions. Overall, the volcanic perturbation of the total ozone column lasts until the end of 1995. Some small artefacts at the poles in 1996 and 1997 arise from the used background aerosol in the NOVOL simulation (see section 2.2).

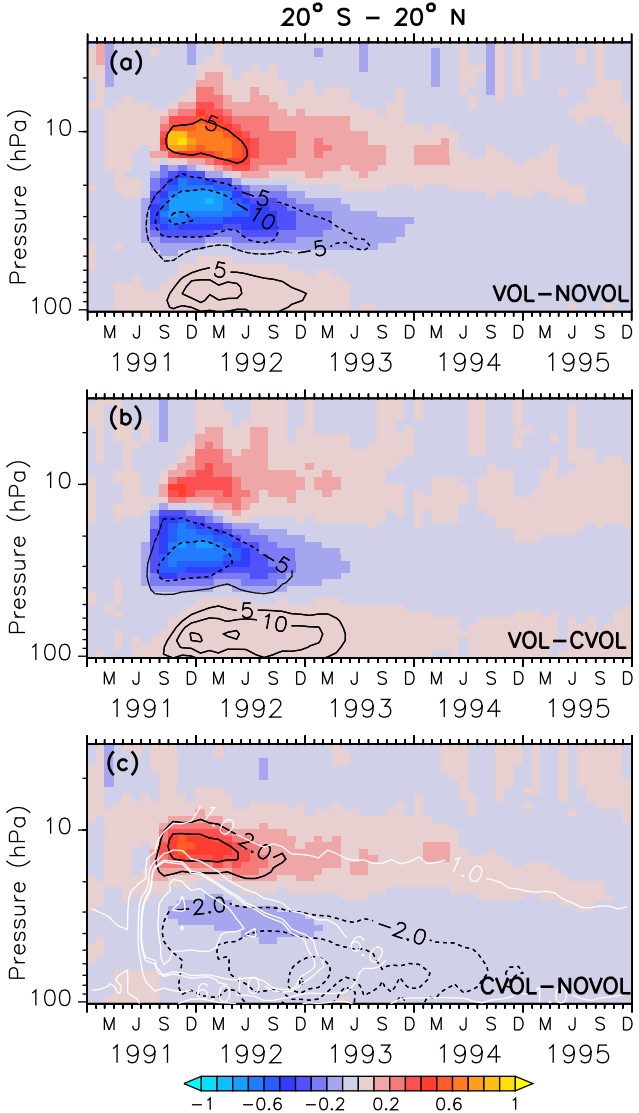

**Figure 7.** Zonally averaged differences of ozone mixing ratios (ppmv) between two simulation results, where **(a)** shows the combined effect (VOL-NOVOL), **(b)** the heating effect (VOL-CVOL), and **(c)** the chemical effect CVOL-NOVOL). Black contours indicate are the corresponding relative changes (%). Shown are averages in the tropics between 20° S and 20° N. Contour intervals are 5 % in the upper and central panel, and 2 % in the bottom panel. White contours in **(c)** show the liquid aerosol surface area density ($10^{-8}$ cm$^2$cm$^{-3}$) in VOL and CVOL with intervals of 1, 5 and 10.

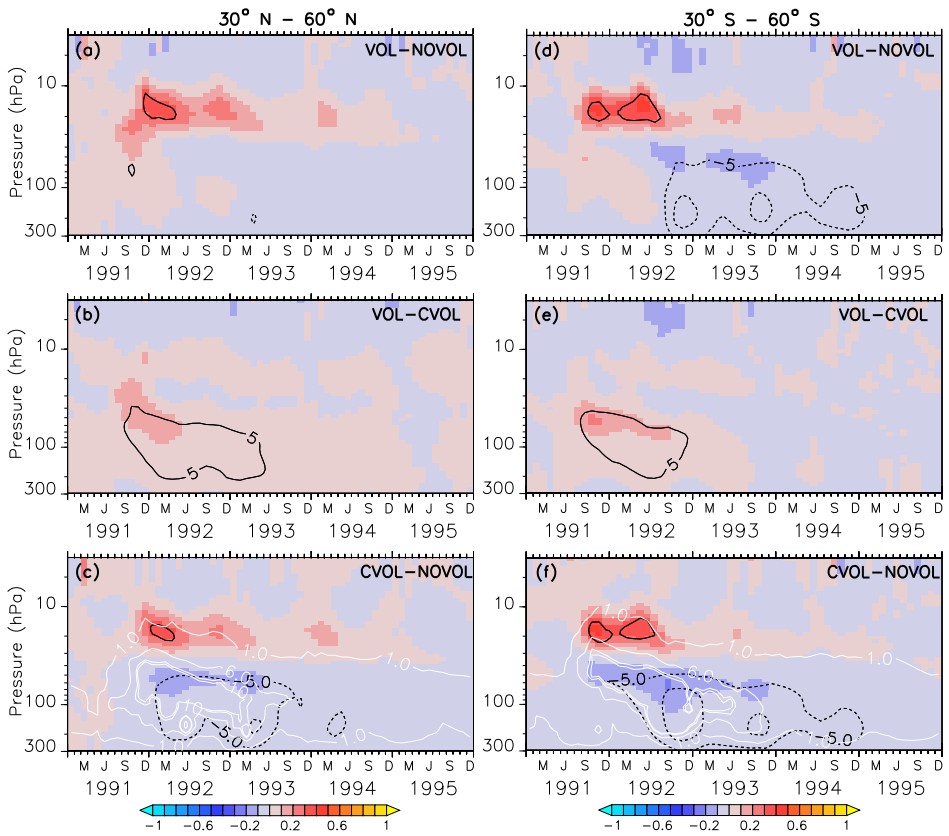

**Figure 8.** Zonally averaged differences of ozone mixing ratios (ppmv) between the simulations VOL - NOVOL **(a)** and **(d)**, VOL-CVOL **(b)** and **(e)** and CVOL-NOVOL **(c)** and **(f)**. Contours indicate the corresponding relative changes (%). Shown are averages over the mid latitudes 30° N - 60° N (left) and 30° S - 60° S (right). Contour intervals are 5 %. White contours in **(c)** show the liquid aerosol surface area density ($10^{-8}$ cm$^2$cm$^{-3}$) in VOL and CVOL with intervals of 1,5 and 10.

## 4.3 Vertical Ozone Distribution

The combined effect on ozone in the tropics is characterised by an increase of 5 % around 10 hPa and by a decrease of 15 % between 20-50 hPa (Figure 7a). This dipole structure appears in both, the heating (VOL-CVOL) and the chemical effect (CVOL-NOVOL), and they sum up to the combined effect (VOL-NOVOL, Kilian, 2018). The largest impact in the middle

5    stratosphere is mostly attributed to the volcanic heating (Figure 7b). The perturbation of ozone lasts about 2 years. Volcanic heating by the aerosol modifies ozone via an increase of the vertical ascent by 20 % and subsequent transport to higher latitudes, which reduces the tropical ozone column (McCormick et al., 1995b; Dameris et al., 2005). The ozone increase at 10 hPa arises partly from the chemical effect (Figure 7c). As it will be more precisely explained in section 4.4, larger liquid aerosol surface modifies the heterogeneous reactions and NO$_x$ is converted into HNO$_3$. Thus, less NO$_x$ is available and the catalytic ozone

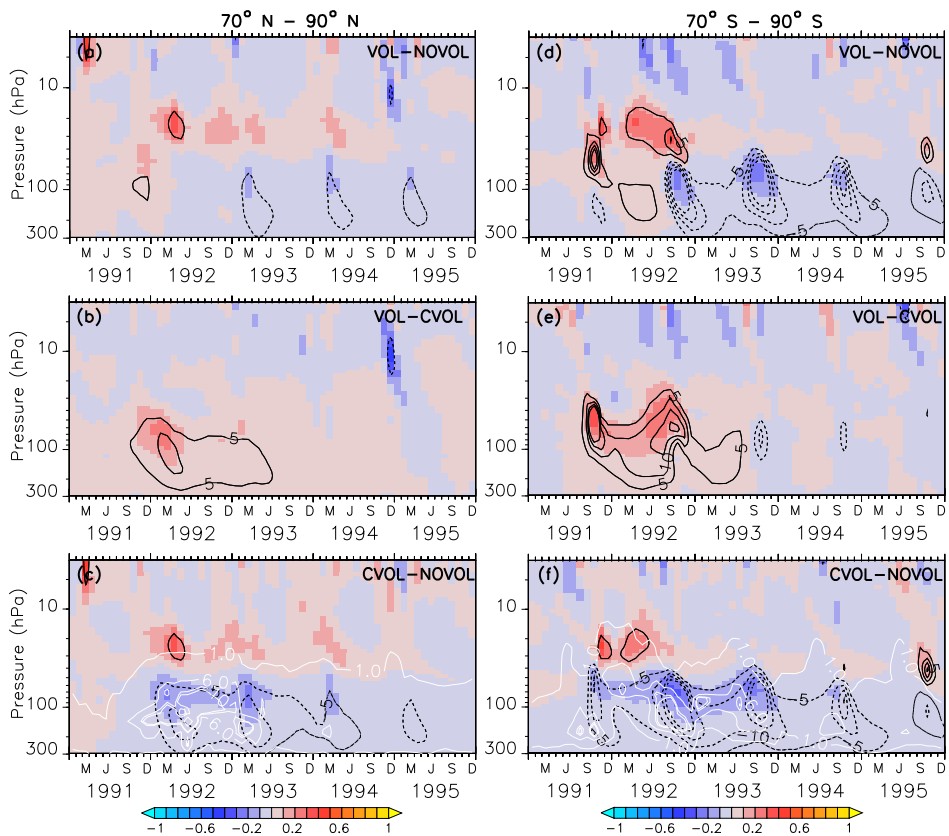

**Figure 9.** Zonally averaged differences of ozone mixing ratios (ppmv) between the simulations VOL - NOVOL **(a)** and **(d)**, VOL-CVOL **(b)** and **(e)** and CVOL-NOVOL **(c)** and **(f)**. Contours indicate the corresponding relative changes (%). Shown are averages over the mid latitudes 70° N - 90° N (left) and 70° S - 90° S (right). Contour intervals are 5 %. White contours in **(c)** show the liquid aerosol surface area density ($10^{-8}$ cm$^2$cm$^{-3}$) in VOL and CVOL with intervals of 1, 5 and 10.

depletion via $NO_x$ slows down. In accordance with Telford et al. (2009b), the chemical ozone decrease in the tropics is smaller than the dynamical effect, due to increased vertical transport (or heating effect).

In the extratropics, the combined $O_3$ perturbation also exhibits a dipole structure with an increase of $O_3$ between 10-30 hPa by up to 5 %, and a small decrease of ozone beneath, which lasts more than one year (Figure 8a). This increase mostly arises

5 from the chemical effect. Again, the $NO_x$ cycle slows down so that less $O_3$ is depleted (see section 4.4). Interestingly, the heating effect and the chemical effect have counteracting influences on the ozone mixing ratio in the lower stratosphere, with an increase due to the volcanic heating, and a decrease in the same order of magnitude by the chemical effect (Figure 8b,c). A similar dipole pattern is also observed in the polar regions, where the perturbation of ozone is even larger than in the extratropics (Figure 9). Similarly, the heating and chemical changes are counteracting with up to +/- 10 % at the North (Figures 9(**b**) and

10 (**c**)) and up to +/- 20 % at the South Pole ((**e**) and (**f**)), mostly between 100 and 300 hPa. In the polar regions the seasonal cycle of the ozone perturbation is large and enhanced by the formation of PSCs (section 4.4.1), so that the strongest effect appears

during the respective spring season. The ozone decrease due to the chemical effect between 100-300 hPa lasts 4 years, whereas the ozone increase by the heating effect in the lower and middle stratosphere already disappears after one year. The next section describes, whether the chemical effect on ozone arises from a perturbation of the ozone production or the ozone loss, and how the net ozone production is affected.

## 4.4  Ozone Budget of the Chemical Effect

In the following, we analyse the chemical effect (CVOL-NOVOL) on the $O_3$ budget for the tropics, the mid-latitudes and the polar regions (Figures 10 - 12). All presented results show the differences between CVOL and NOVOL of the $O_3$ production and loss rates given in mol/s. The shown vertical levels are selected according the largest perturbation of the $O_3$ mixing ratio (Figure 7). The $O_3$ loss rates are determined by different contributions of the catalytic cycles of $NO_x$, $ClO_x$, $HO_x$, $O_x$ and $BrO_x$.

The increase of the $O_3$ mixing ratio in the tropics at 12 hPa (section 4.3) is explained by a strong decrease of the $O_3$ loss rate (Figure 10a). The $O_3$ gain is mostly compensated by a reduced $O_3$ production. The decrease of the $O_3$ production at 12 hPa is caused by an increase of the chemical composition above, especially $O_3$, which dims the solar radiation and thus slows down the photolysis of oxygen and reduces the production of $O_3$ (Kilian, 2018). The strong reduction of the $O_3$ loss rate (Figure 10a) arises from a less active $NO_x$ cycle (Figure 10b), because $NO_x$ is transformed into the reservoir gas $HNO_3$ due to the enhanced liquid aerosol surface. On the other hand, $HO_x$, $ClO_x$ and $O_x$ cycles accelerate and thus partly counteract the $NO_x$ cycle slow down towards a new chemical equilibrium. This leads to the increase of the $O_3$ mixing ratios at 12 hPa. The duration of the disturbed ozone loss rates (Figure 10b) reflects the presence of the upper part of the volcanic aerosol cloud. At 30 hPa $O_3$ slightly decreases, because the $HO_x$, $ClO_x$ and $O_x$ cycle accelerate more than the $NO_x$ cycle decelerates (Figure 10c).

In the mid-latitudes of the northern hemisphere the strongest $O_3$ increase appears at 20 hPa. Again, this is caused by a change of the ozone loss and production due to the volcanic aerosol, although its order of magnitude is smaller than in the tropics (Figure 11a). However, the perturbation lasts longer. The differences of the production and loss rates in the southern mid-latitudes (Figure 11b) are half as large as in the tropics, but their duration is similar. The maxima in ozone loss rates correspond to the upper height limit of the volcanic cloud. The volcanic perturbation of the ozone budget lasts until spring 1993 on the northern hemisphere, and until summer 1992 on the southern hemisphere (Figure 11e,f). In each case, the decrease of the ozone loss rate can be attributed to a less active $NO_x$ cycle (Figure 11e,f) due to conversion of reactive $NO_x$ on the aerosol surface into $HNO_3$. Note that at this height the $NO_x$ cycle dominates the ozone depleting process.

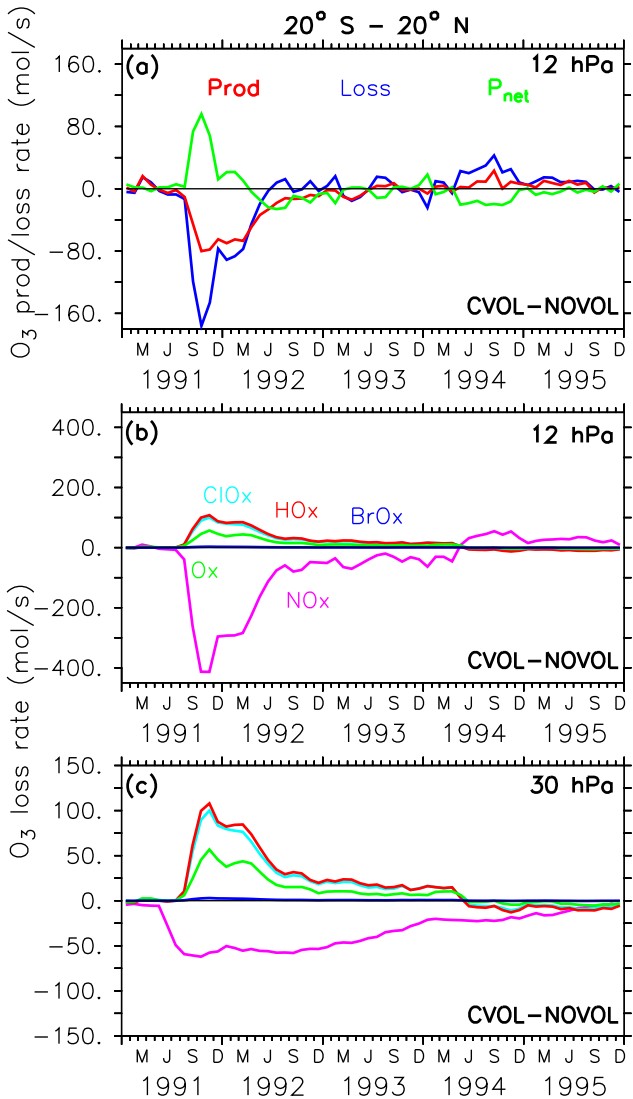

**Figure 10.** Differences (CVOL-NOVOL), i.e. chemical effect of ozone production (Prod) and loss (Loss) rates and corresponding net ozone production ($P_{net}$ = Prod - Loss) **(a)** at 12 hPa. Differences (CVOL-NOVOL) of the ozone loss rates (mol/s) for each catalytic ozone loss cycle of CVOL-NOVOL at 12 hPa **(b)** and 30 hPa **(c)**. The shown level (12 hPa) is located in the maximum of the volcanic ozone perturbation. Shown are zonally averaged and latitudinally (20° S and 20° N) summed rates in mol/s.

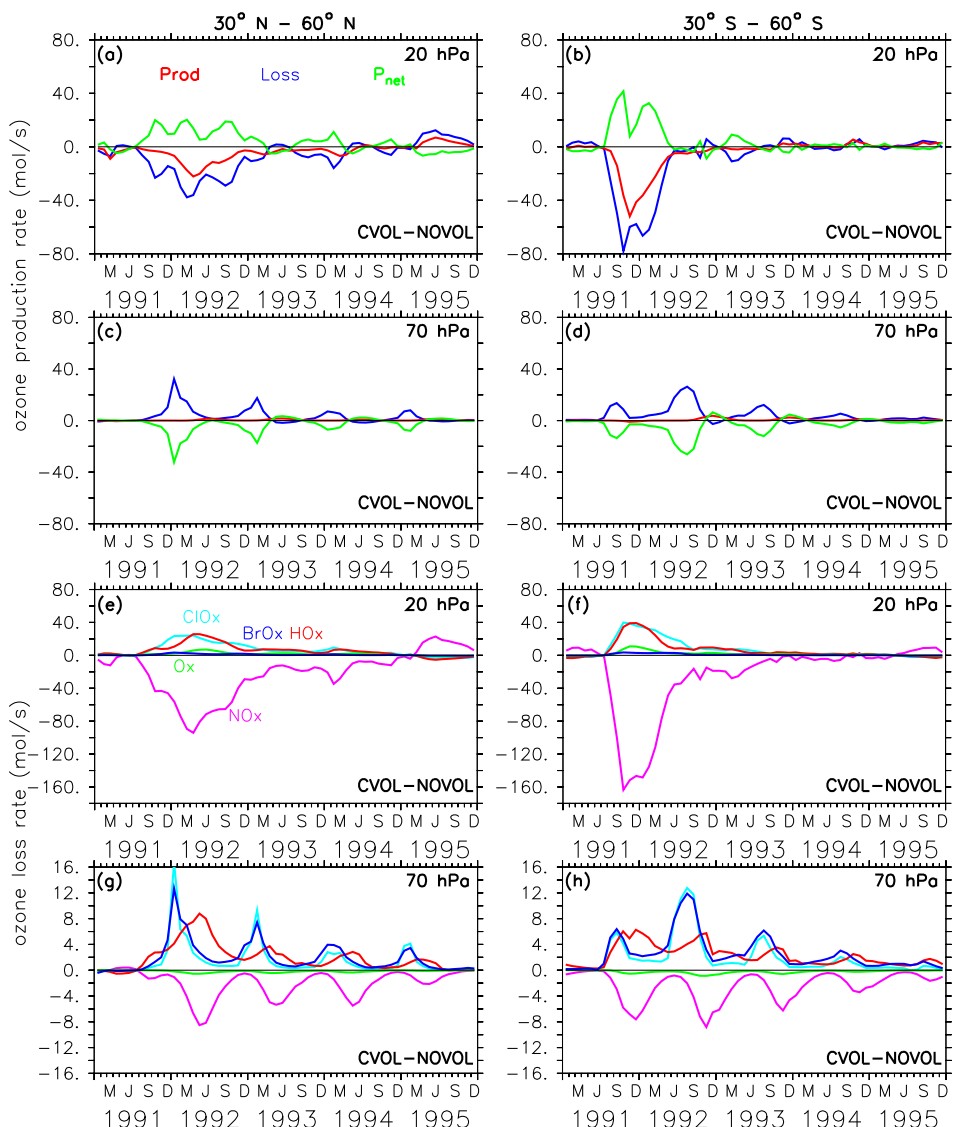

**Figure 11.** Chemical effect (CVOL-NOVOL) of ozone production (Prod) and loss (Loss) rates **(a)**-**(d)** in mid-latitudes and corresponding net ozone production ($P_{net}$ = Prod - Loss). **(a)** (20 hPa) and **(c)** (70 hPa) show zonally averaged and latitudinally (30° N and 60° N) summed rates in mol/s. **(b)** and **(d)**: similar as for **(a)** and **(c)**, but for 30° S-60° S. The shown levels are located in the maximum of the volcanic ozone perturbation. Chemical effect (CVOL-NOVOL) of the zonally averaged and latitudinally summed ozone loss rates (mol/s) for each catalytic ozone loss cycle **(e)**-**(h)** for the northern hemisphere at 20 hPa **(e)** and 70 hPa **(g)**. **(f)** and **(h)** are similar to **(e)** and **(g)**, but for 30° S-60° S, respectively.

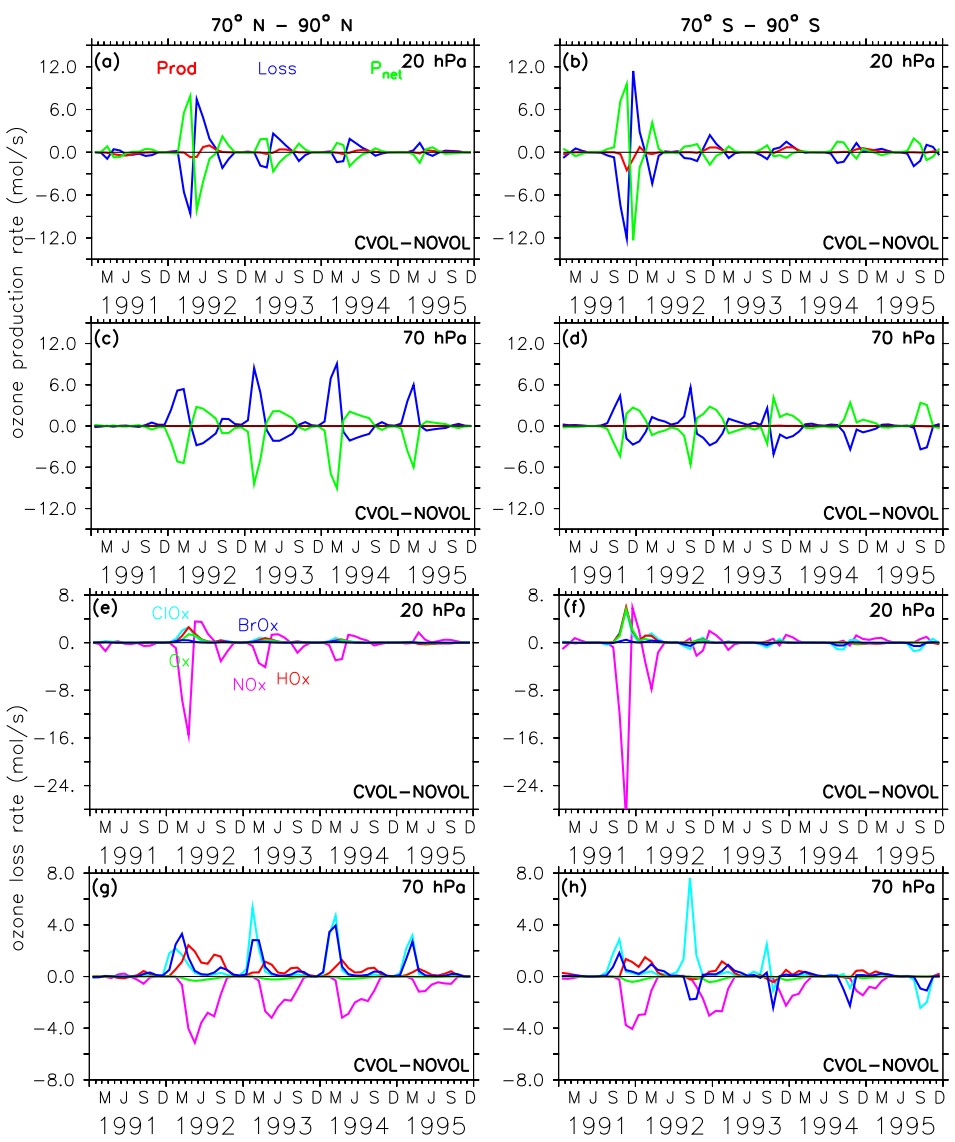

**Figure 12.** Chemical effect (CVOL-NOVOL) of ozone production (Prod) and loss (Loss) rates **(a)**-**(d)** in polar regions and corresponding net ozone production ($P_{net}$ = Prod - Loss). **(a)** (20 hPa) and **(c)** (70 hPa) show zonally averaged and latitudinally (70° N and 90° N) summed rates in mol/s. **(b)** and **(d)**: similar as for **(a)** and **(c)**, but for 70° S-90° S. The shown levels are located in the maximum of the volcanic ozone perturbation. Chemical effect (CVOL-NOVOL) of the zonally averaged and latitudinally summed ozone loss rates (mol/s) for each catalytic ozone loss cycle **(e)**-**(h)** for the polar region at 20 hPa **(e)** and 70 hPa **(g)**. **(f)** and **(h)** are similar to **(e)** and **(g)**, but for 70° S-90° S, respectively.

In the mid-latitudes at 70 hPa the negative differences in the net ozone production (Figure 11g,h) arise solely from an increase in ozone loss rates and lead to the observed $O_3$ decrease in the lower stratosphere in the mid-latitudes. Moreover, a strong seasonal cycle with maximum loss rates in the respective winter is apparent. The seasonal increase in ozone loss rates is related to the predominance of the $ClO_x$ and $BrO_x$ cycle during the respective winter seasons. The presence of volcanic aerosols lead to a conversion of $NO_x$ into the reservoir $HNO_3$. Moreover, during the respective winter months with only minor sunlight, $NO_x$ remains in the reservoir gas $HNO_3$. The reduction of $NO_x$ accelerates the $ClO_x$ and $BrO_x$ cycle in late winter of 1992 and 1993, where the $ClO_x$ and $BrO_x$ cycles reach there maxima in depleting $O_3$ (Figure 11g,h). The increase of the ozone loss rate due to the halogens is not counterbalanced by any other cycle, which explains the strongest decrease of ozone at 70 hPa in late winter of 1992 and 1993 (Figure 8b). Volcanic aerosols plus more water vapour (more OH) in summer increase the importance of the $HO_x$ cycle, which dominates the $O_3$ removal below 20 km (50 hPa, Kilian 2018, Figure 4.10). Together with more $NO_x$ removal through the volcanic cloud, the relative importance of the $HO_x$ cycle increases.

In the polar regions the ozone loss rates at 20 hPa also show a seasonal cycle with a strong decrease in the first respective spring season after the eruption, but followed by an abrupt increase in the following summer season (Figure 12a,b). The rapid decrease of the ozone loss rate, which is related to the first occurrence of the volcanic aerosol cloud in the north polar region in spring 1992 is 10 times smaller than in the mid-latitudes (Figure 12a,b and 12e,f). This abrupt change of sign is related to the beginning of solar insolation after the polar night. The strong reduction of $O_3$ loss rates are related to the decrease in the $NO_x$ cycle in summer, that is not compensated by the $HO_x$ cycle as in the mid-latitudes (Figure 12e,f). At 20 hPa, and only in the first spring season after the eruption, and only in the south polar region, the $O_x$ cycle becomes stronger in ozone depletion. The $HO_x$ cycle shows a local minimum during the summer month (June and December).

The polar ozone decrease at 70 hPa is caused by an increase of the $O_3$ loss rate (Figure 12c,d). The responsible catalytic cycles have a similar seasonal variability as in the mid-latitudes (Figure 12g,h). Again, the efficiency of the $NO_x$ cycle decreases, which is halfway compensated by the $HO_x$ cycle.

This can be explained by the gas-phase reaction $HO_2 + NO \longrightarrow OH + NO_2$, which affects the interconversion between $HO_2$ and OH (Seinfeld and Pandis, 1998). Their ratio is controlled by the temperature dependent reaction rate and by the concentrations of $O_3$ and NO. In the lower stratosphere the reaction $HO_2 + O$ is negligible, and $HO_2 + O_3$ is predominant. Since $NO_x$ decreases in the lower stratosphere starting in spring 1992, more $HO_2$ is available to react with $O_3$ instead of with NO. This leads to an increase of the meaning of the $HO_x$ cycle for the lower stratosphere.

The acceleration of the $ClO_x$ and $BrO_x$ cycles in springtime starts in 1992 especially at the North Pole. Note, that except for the first year after the eruption at the South Pole the $BrO_x$ cycle is out of phase with the $ClO_x$ cycle at 70 hPa. Probably, the deactivation of chlorine (Cl) and bromine (Br) atoms by a stronger polar vortex during the polar night is enhanced by the volcanic aerosol, and accelerates both cycles in spring.

### 4.4.1 Impact of the Polar Stratospheric Clouds

Next, the effects of the volcanic aerosol on the formation of PSCs are investigated. PSCs accelerate the heterogeneous chemistry, because fast chemical reactions take place on the PSC particles. During the polar night the strong isolated polar vortex

prevents bonded chlorine atoms to deplete ozone. PSCs appear during the polar night at heights of 25-80 hPa, that is exactly where the largest increase of the aerosol surface occured. As soon as the first sunrays appear, $Cl_2$ is photolysed and Cl atoms destruct ozone. PSCs amplify the ozone destruction, because more particle surfaces are available for heterogeneous reactions. More ozone destructive reactions occur at the surface of the PSC particles to form $Cl_2$ and $Br_2$. The volcanic heating increases

5    the stratospheric temperature between 30-70 hPa by 3 K in the polar regions, implying that less PSCs form. Therefore, the heating effect (VOL-CVOL) shows less PSCs in both polar hemispheres. However, the chemical effect shows a temperature reduction in the lower stratosphere due to less ozone (Figure 9c,f), resulting in an enhanced formation of PSCs at the North and the South Pole (Figure 13, CVOL-NOVOL). At the North Pole, the PSC covered area increases between November 1991 and March 1992 (Figure 13a, CVOL-NOVOL). At the South Pole the increase of the PSC area appears in the springtime, as well as

10   in the autumn, when the first PSCs are formed (Figure 13b, CVOL-NOVOL). Although we cannot quantitatively differentiate between the direct effect from PSC formation and the ozone destruction, it is likely that the latter contributes to the ozone depletion in the respective spring season by enhancing the $ClO_x$ cycle at 70 hPa.

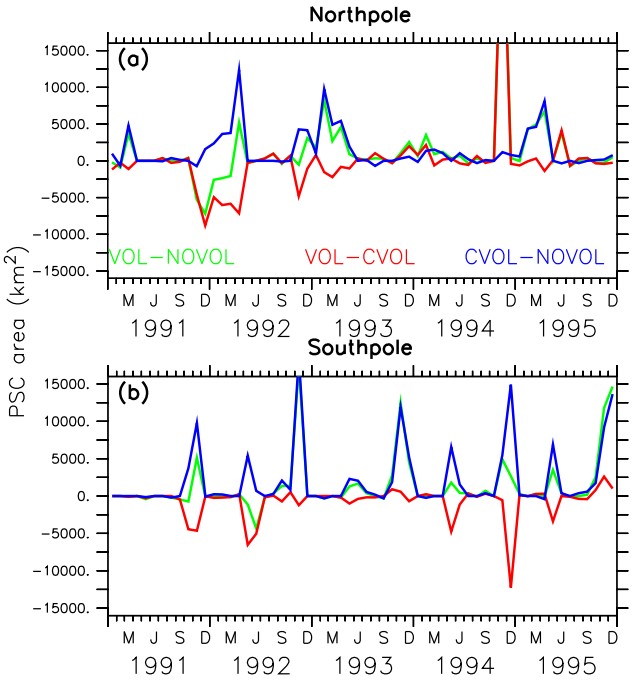

**Figure 13.** Differences of polar stratospheric cloud (PSC) covered area (km$^2$) at the North Pole (60° N - 90° N) **(a)** and South Pole (60° S - 90° S) **(b)** for the combined effect (green), the heating effect (red) and the chemical effect (blue).

## 4.5 Stratospheric Water Vapour

Similar as in Löffler et al. (2016), we find that additional water vapour is transported into the tropical stratosphere, due to the heating of the tropopause by the volcanic cloud. The separation of the heating and the chemical effect shows that only the heating effect contributes to the increase of SWV (Figure 14a).

5     In the case of the chemical effect, which has not been considered by Löffler et al. (2016), the lower stratosphere is cooled by 0.4 K (due to less ozone) and hence less water vapour enters the stratosphere (Figure 14b). Moreover, in September 1991 a negative SWV perturbation (outside the tape recorder) appears, following the volcanic plume (Figure 14b) in the first year after the eruption. This pattern is caused by sublimation of SWV into liquid and ice on the volcanic condensation nuclei.

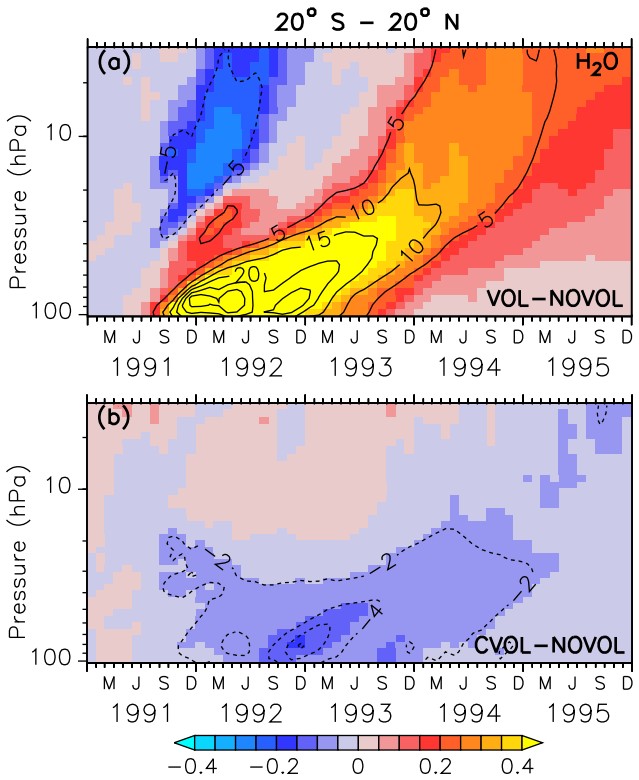

**Figure 14.** Zonally averaged differences of stratospheric water vapour (ppmv) between the simulations, where **(a)** shows the combined effect (VOL-NOVOL) and **(b)** the chemical effect (CVOL-NOVOL). The black contours show the relative change (%) of water vapour in comparison to NOVOL and CVOL, respectively. Shown are tropical averages (20° S - 20° N). Contour intervals are 5 % **(a)** and 2 % **(b)**.

## 4.6 Methane

10   Stratospheric methane ($CH_4$) is affected by the aerosols mainly due to the heating effect. Already 3 months after the eruption an increase of methane between 40 hPa and beyond 10 hPa can be noticed with changes of more than 80 ppbv, corresponding

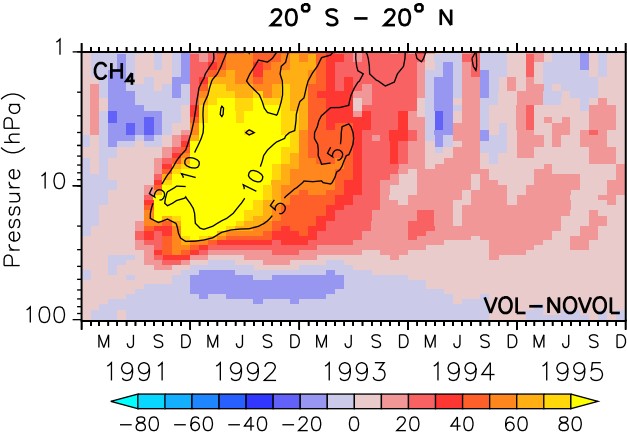

**Figure 15.** Zonally averaged differences of the methane mixing ratio (ppbv) for the combined effect (VOL-NOVOL). The black contours show the relative change (%) of methane in comparison to NOVOL. Shown are tropical averages (20° S - 20° N). Contour intervals are 5 %.

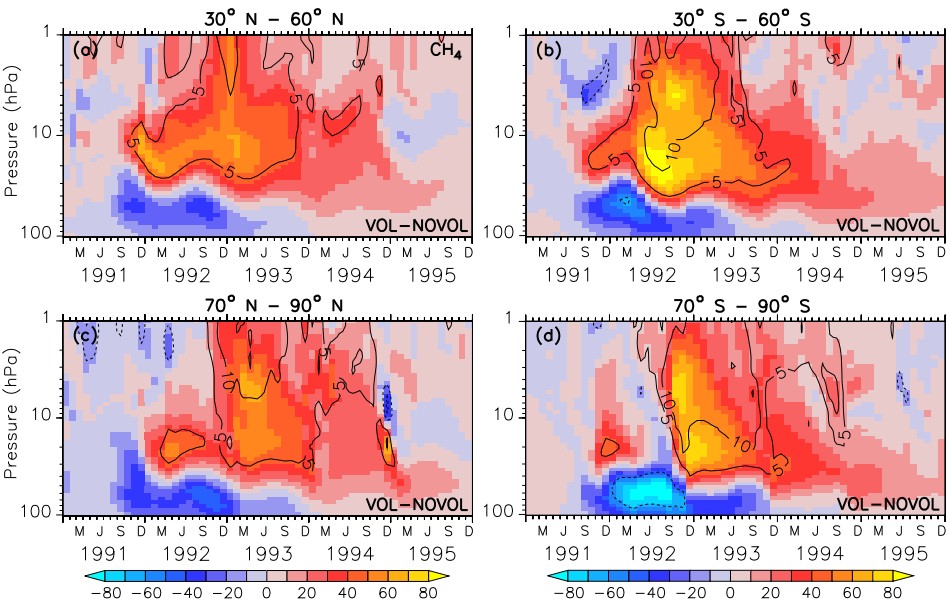

**Figure 16.** Zonally averaged differences of methane (ppbv) of the combined effect (VOL-NOVOL). The black contours show the relative change (%) of methane in VOL comparison to NOVOL, respectively. Shown are extratropical averages (**(a)** and **(b)**: 30° N - 60° N and 30° S - 60° S; **(c)** and **(d)** : 70° N - 90° N and 70° S - 90° S). Contour intervals are 5 %.

to a change of 10 % with respect to the absolute $CH_4$ in the unperturbed simulation NOVOL (Figure 15a). The upward propagating elevated values of $CH_4$ are the result of the enhanced tropical vertical ascent, which leads to a transport from relative methane rich air in the lower stratosphere into upper levels. The surplus of $CH_4$ is not depleted, which is an indicator that the stratospheric lifetime of $CH_4$ increased. In the lower stratosphere we find less $CH_4$ due to the volcanic eruption. The

dipol pattern of the $CH_4$ perturbation spreads with time into the mid-latitudes (Figure 16a,b) and into the polar regions (Figure 16c,d). The increase of $CH_4$ in the tropical regions is partly transported by the deep branch of the BDC within a year to higher latitudes. But there is also a decrease of $CH_4$ partly transported by the lower branch of the BDC reaching mid-latitudes within the period of 1.5 - 3.5 years after the eruption.

## 5  Discussion

The differences of the simulations (VOL-CVOL) separate the heating, and (CVOL-NOVOL) the chemical effect. In particular, we investigated the long- and shortwave heating effect of the volcanic aerosol on the temperature and transport, as well as on ozone, SWV and $CH_4$. The ozone development of the simulation VOL after the eruption is in agreement with the satellite based SWOOSH data (Davis et al., 2016) and the TOMS data (TOMS-Science-Team, 2004). We perform specified dynamics simulations with a prescribed aerosol distribution (CCMI dataset; see sec. 2.1). Thus, we miss the primary feedback of the aerosol induced heating on the transport of the aerosol. But since we are interested more in the chemical effects here, this choice is advantageous, because the aerosol distribution is derived from observations. The definition of a "background" aerosol distribution, relevant for the NOVOL simulation refers to a period, when volcanic activity and related aerosols are less present. Nevertheless, in the year 2011, which we selected for a background aerosol distribution, the medium size volcanic eruption of Nabro occurred and thus is also included in the CCMI dataset. In comparison with other eruptions Nabro emitted only 1 Tg $SO_2$ (Pinatubo: 20 Tg) into the atmosphere up to heights of 18 km (Bourassa et al., 2012). The tropopause height at 15° N is at 18 km and thus in the region of the upper part of the volcanic plume. The eruption occurred during the months of June-September 2011 (and is therefore repeated every year in the background aerosol). An important amount of aerosol is only present in the mid to high latitudes (not shown). Ozone depletion due to Mt. Pinatubo might be underestimated in high latitudes (Figure 6c) in our analysis. For the Arctic region, the ozone depletion declines systematically already in June, the month, where aerosols of Nabro are present in NOVOL.

Using nudged simulations has several advantages over free-running simulations to study the impact on the chemistry: The temperature response is closer to observations, which is important, as ozone chemistry is temperature dependent. The results appear less noisy. Our nudged simulations (VOL, CVOL and NOVOL) are similar with respect to the synoptic situation, thus the effect of aerosol heating on subgrid-scale chemistry and transport of ozone can be contoured more clearly. This would be more difficult, if one allows the synoptic situation to evolve freely. Moreover, for such an approach a large set of ensemble simulations is necessary (see for instance Aquila et al., 2013). Another possibility is to perform simulations, in which nudging is applied up to 100 hPa, i.e. below the region where the aerosol heating occurs, together with a weak QBO nudging, as this avoids the noisy and expensive free running ensembles. Nudging (to ERA-interim reanalysis data) is applied to the prognostic variables temperature, divergence and vorticity, and the logarithm of the surface pressure. We apply this nudging in the spectral space by omitting the nudging of wave-zero of the temperature, thus we do not correct temperature biases, implying that the absolute temperature can evolve. Moreover, the nudging is applied as low-normal mode insertion, i.e. down to the synoptic scale only, with comparably long relaxation times. The nudging is applied such that the large(r than synoptic) scale patterns

correspond to those of ERA-Interim, but not the absolute temperature. This means that the synoptic situation is that of ERA-Interim, whereas sub-synoptic variations can evolve freely, as for instance the influence of the volcanic cloud on the vertical velocity and the temperature profile. SWV, strengthening of the vertical motion, and polar stratospheric clouds are influenced by nudging only indirectly. Because we nudge all three simulations with ERA-Interim data and prescribe SSTs/SICs, the SST part of the volcanic signal is included in all simulations, and cancels out in the differences calculations. Therefore, our approach isolates the radiative and chemical effect of the volcanic aerosol on the ozone distribution.

The simulated volcanic heating of 3.5-4 K between 50-60 hPa in VOL agrees well with results from Labitzke and McCormick (1992), who observed a stratospheric heating at 50 hPa of 3-4.5 K in the tropics. Angell (1997a), who used radiosonde data, figured out a temperature increase of approximately 3-4 K between 30-50 hPa in late 1991. More recent studies (Revell et al., 2017; Kuchař et al., 2017) stated, that simulations using CCMI aerosol data tend to overestimate the temperature response in the stratosphere. And this in turn may effect the amount of water vapour entering the stratosphere at warmer simulated cold point temperatures than observed. The results of Löffler et al. (2016) point in a similar direction. Their nudged simulation using CCMI aerosol data also tend to overestimate the temperature anomaly after the Mt. Pinatubo eruption. Grant et al. (1992), Schoeberl et al. (1993b) and Angell (1997a) investigated total ozone column depletions of 5-8 % in the tropical region 6 months after the Mt. Pinatubo eruption. Our findings with up to 6 % column loss agree with these results and with measurements from the Mauna Loa observatory on Hawaii, showing a 5 % $O_3$ column loss, quite well (Randel et al., 1995). McCormick et al. (1995a) indicated the same order of magnitude from observation data with an ozone loss of 6-8 % over the Equator in the first several months after eruption. Grant et al. (1992) reported the absolute change of the ozone column in the tropics to be 13-20 DU, which is in agreement with our study with decreases of 10-18 DU. Grant et al. (1992) and McCormick et al. (1995a) observed the largest tropical loss of ozone with 20 % between 24-25 km (25-30 hPa level), which is 5 % more than the perturbation in our study. Although the ozone production rates slow down (due to the local increase in ozone abundance), they might still be too high, because the simulated photolysis rates do not account for the presence of the volcanic aerosol. The ozone change after a volcanic eruption is a combination of an ozone decrease of the vertical column in the tropics directly after the eruption, and ozone increase in the mid latitudes and polar regions in the first year after the eruption. Poberaj et al. (2011) reported a similar temporal behavior of the ozone decrease after the Mt. Pinatubo eruption. They quantified an ozone decrease of 3-4 % in the southern mid-latitudes. Telford et al. (2009b) reported a total ozone reduction of 10 DU averaged between 30-60° N and 12 DU between 30-60° S caused by the chemical effect. In general, the chemical effect on ozone perturbation becomes more important at higher latitudes (Telford et al., 2009b). Our results show that the the ozone increase in the extratropics in the year after the eruption can be solely attributed to the heating effect, i.e. the effect of increased transport of ozone with the BDC. Joshi and Shine (2003) simulated water vapour increases of 20-30 % in the tropics between 10° S - 10° N after the Mt. Pinatubo eruption. Considine et al. (2001) simulated SWV and $CH_4$ anomalies between 0-10° N and showed increases of the vertical ascent in the tropical lower stratosphere of up to 24 %. Referring to their Figure 16b, they stated a positive SWV anomaly of 15-20 % starting in October 1991 following the tape recorder signal. Our study confirms these results quantitatively, and in terms of spatial and quantitative distribution. Furthermore, Considine et al. (2001) found increased values of $CH_4$ with 2 % at 10 hPa, which reached amplitudes of up to 10 % at 0.1 hPa. In our simulation, the largest relative methane

anomalies caused by volcanic heating occur between 20 hPa and up to 1 hPa. Probably, this methane increase is predominantly caused by the stronger vertical ascent and less by secondary chemical effects. We cannot explicitly differentiate in the heating effect between a stronger vertical transport and a smaller extent temperature influence on the chemistry. Winterstein et al. (2019) showed that the concentration of OH, which determines the lifetime of $CH_4$ is dependent of SWV and $O_3$. This leads us to the limitation of our study, namely that the perturbation of the chemical composition arises from a temperature induced enhancement of the transport and of the temperature dependent heterogeneous reaction rates (secondary effect). Both effects are included in the heating effect (VOL-CVOL), but can not be separated in this study.

## 6  Summary

We presented an analysis with the focus on the changes of ozone, methane and water vapour, caused by the volcanic aerosols of the Mt. Pinatubo eruption in 1991 as simulated with the chemistry-climate model EMAC. We performed three specified dynamics simulations with prescribed volcanic aerosol, which differ with respect to the interaction between volcanic aerosol, chemistry and radiation: VOL (volcanic aerosol: interaction with radiation AND heterogeneous chemistry), NOVOL (no volcanic aerosol) and CVOL (volcanic aerosol: interaction with chemistry, NO interaction with radiation). This enables the separation of the effect of volcanic heating on the atmospheric composition from the chemical effect, i.e. the effect of heterogeneous reaction rates on volcanic aerosols on the ozone distribution.

Although we use specified dynamics simulations, the effect of volcanic heating on the BDC is visible in an increase in vertical velocity, especially in the tropics, with subsequent increased horizontal transport of heat and tracers with the BDC.

Further, the use of specified dynamics simulations allows us to eliminate the effect of the large-scale meteorology (e.g. QBO, ENSO) when the simulations are compared. This is important, as for instance the eruption of the Mt. Pinatubo coincided with a time period of strong upwelling in the tropics with related cooling. As a result, the observed tropopause temperatures were lower than expected through the eruption. This might also be the cause, why free-running models have a tendency to overestimate the temperature response in the stratosphere. This is the fundamental difference to other studies on the effect of the Mt. Pinatubo eruption on the ozone distribution, as by Aquila et al. (2013) and by Muthers et al. (2015). They studied, among others, the contribution of interannual dynamical variability on the ozone distribution after the eruption.

The major finding in our study is that the chemical effect shows up globally as a dipole structure in the vertical ozone distribution with an ozone increase around 10-30 hPa and a decrease below down to the tropopause. These two vertical regions are characterized by a different importance of the $NO_x$ cycle.

The dominant chemical increase in ozone (1 year after the eruption around 10-30 hPa) is caused by a reduction of $NO_x$ into $HNO_3$ on the volcanic aerosol, i.e. a slow down in the ozone depleting $NO_x$ cycle.

The ozone decrease (in the lower stratosphere in mid- to high latitudes), however, is caused by an acceleration of the ozone depleting cycles $ClO_x$, $BrO_x$ and $HO_x$ as a result of the reduced $NO_x$ cycle. Moreover, these chemical cycles show a strong seasonal variablity in mid and high latitudes. SWV and $CH_4$ were increased by the heating effect by up to 25 % and 10 % in the tropics, respectively, and in contrast to $O_3$, for SWV and $CH_4$ the chemical effect has less importance.

*Acknowledgements.* This study was funded by the DLR project KliSAW. We acknowledge the use of the tool cdo (https://code.zmaw.de/projects/cdo) for the processing of data and the program Ferret from NOAA's Pacific Marine Environmental Laboratory (http://ferret.pmel.noaa.gov) for analysis and creation of graphics in this study. We thank Davis et al. (2016) for providing the SWOOSH data. We gratefully acknowledge the support of NASA/GSFC's Ozone Processing Team in providing the TOMS data presented

5    here. The SBUV/2 data were obtained from NOAA/NESDIS with support from the NOAA Climate and Global Change Atmospheric Chemistry Element (Wellemeyer et al., 2004). We further thank the German Climate Computing Centre (DKRZ) for providing computational resources for the simulations, as well as for data processing and analysis. We thank Heidi Huntrieser and Volker Grewe for the internal review of our manuscript.

**Data and Code availability**

10    The simulation results and the computer code used here are archived at the German Climate Computing Center (DKRZ) and are available on request.

**Author Contributions**

MK analysed the data and wrote the manuscript. SB and PJ designed the experiments, performed the model simulations and contributed to the manuscript.

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
