# Peer review of "Impact of the eruption of Mt. Pinatubo on the chemical composition of the stratosphere"

_Atmospheric Chemistry and Physics, 2020_

## Short Comment (SC1) · 6 Apr 2020

Valentina Aquila

aquila@american.edu

The results of our paper (Aquila et al. 2013) are misrepresented at page 10 line 5 ("This result is in contradiction to Poberaj et al. (2011) and Aquila et al. (2013), who attributed the absence of ozone depletion in the southern hemisphere to interannual dynamic variability.") Aquila et al. (2013) showed that the absence of ozone depletion in the southern hemisphere was caused by the stronger Brewer-Dobson circulation due to the volcanic heating. From our abstract: "The authors' simulations show that the heating due to the volcanic aerosol enhanced both the tropical upwelling and Southern Hemisphere extratropical downwelling. This enhanced extratropical downwelling, combined with the time of the eruption relative to the phase of the Brewer–Dobson circulation, increased Southern Hemisphere ozone via advection, counteracting the

ozone depletion due to heterogeneous chemistry on the Pinatubo aerosol." It is also clearly shown in Fig. 7 of our paper.

If I have understood correctly this manuscript, I also think that the sentence "For the first time, the effects of volcanic heating and heterogeneous chemistry on the chemical composition, caused by the volcanic aerosol, are separated", which is contained in the abstract of Killan et al., is incorrect, as other studies have done that before (among which our Aquila et al., 2013).

Reference Aquila, V., Oman, L. D., Stolarski, R., Douglass, A. R. and Newman, P. A.: The Response of Ozone and Nitrogen Dioxide to the Eruption of Mt. Pinatubo at Southern and Northern Midlatitudes, J. Atmos. Sci., 70(3), 894–900, doi:10.1175/JAS-D-12-0143.1, 2013.

———————————————————

---

## Author Comment (AC1) · 8 Apr 2020

Dear Valentina,

Thank you very much for your remarks.

We apologize for the wrong citation, which we will remove from this place in the revised manuscript. Moreover, our sentence "For the first time, the effects of volcanic heating and heterogeneous chemistry on the chemical composition, caused by the volcanic aerosol, are separated" is in this form indeed incorrect as you mention correctly. The formulation we have chosen is misleading and needs to be precised. Nevertheless, we performed nudged simulations with which the chemical effect is separated from the heating effect for the first time. Our simulations are different from those of Telford et

al. (2009), who indeed also performed nudged simulations, but their chemical effect includes the volcanic heating and thus corresponds to our simulation with the combined effect of volcanic heating and heterogeneous chemistry. We will clarify this in the revised manuscript.

---

## Referee Comment (RC1) · Anonymous Referee #1 · 14 Apr 2020

Review of Kilian et al. acp-2020-147

This is a simple, carefully done study. I like how clearly you have defined your three experiments, and the aims are clear. The scientific conclusions follow well from your experiments and analysis. I am recommending minor revisions.

General Comments

The introduction does a good job of explaining the context of your study and exactly what the knowledge gaps are. However, what was missing is why those knowledge gaps are important to address. More specifically, why is it important to separate the chemical effect of heterogeneous chemistry from the heating effect of the volcanic heating? I think a sentence or two would be useful here.

Since you're using prescribed surface area density, it might be useful to talk about which processes/feedbacks you're missing that might affect your results.

It would be useful to talk about how the nudging may have affected your results. For example, on Page 7 you talk about transport of SWV, increases in vertical motion, and polar stratospheric clouds. Are these things affected by the nudging? Are others?

There are quite a few typos. I've pointed out some but not others. I'd recommend the authors spend some time proofreading.

Specific Comments

Page 2, line 12: Poleward misspelled

Page 5, lines 2-5: These read like throwaway comments. What are the sociopolitical impacts?

Page 5, line 5: Assess misspelled

Page 10, line 1: Is this really exactly linear? I think you need to demonstrate this more clearly.

---

## Referee Comment (RC2) · Anonymous Referee #2 · 28 Apr 2020

**1 General Comments**

The paper tries to separate the effects of aerosol heating and aerosol chemistry on ozone. It is, however, an odd concept to nudge temperatures and winds to observations in the region where temperature and dynamics changes due to aerosol should be analysed (page 5 and 6). This must have consequences for the results. To get some feeling for the introduced artifact it would be good to perform a sensitivity simulation with nudging only up to about 100hPa to get the tropospheric wave forcing but the unperturbed effects of calculated radiative heating due to Pinatubo aerosol on stratospheric dynamics, QBO nudging based on the Singapore data might be on for this case. The manuscript might be published after revision.

**2 Specific comments**

Page 1, line 5: The use of CCMI slang hides the problem with nudged temperatures up to 10hPa.

Page 1, line 10f: Separate between aerosol and ozone heating (see section 4.1), please reword for clarity.

Page 2, line 8: Don't forget to mention the terrestrial infrared.

In the introduction references to earlier studies with EMAC on Pinatubo are missing.

Page 4, line 6: Is the assumed distribution monomodal or how many modes are included?

Page 5, line 17: 2011 is perturbed by the medium size volcanic eruption of Nabro and therefore not background. Where are the data from? SAGE died in 2005.

Page 7, line 18: More details please, NIR, IR. Transport effect on ozone heating?

Page 9, section 4.2: Why is 1993 not addressed? In this year were the largest effects on total ozone in midlatitudes in observations but also in Fig. 5. It would be also good to compare with observations here (e.g. TOMS).

Page 10, line 1: This appears to be in contradiction to Fig. 4.

Page 10, line 31: You may mention PSCs here.

Page 12: There are several ways to separate the catalytic ozone destruction cycles. You may discuss the meaning of $HO_2+NO$ for aerosol perturbed lower stratospheric ozone.

Figures 10 and 11 might be merged, as well as Figs. 12 and 13. It should be also better to show a common pressure level in the lower stratosphere in tropics and extratropics (Figs. 9 to 13) instead of 30 and 20hPa. What is included in $O_x$? Please define, the

Chapman cycle alone cannot explain the curves for the lower stratosphere.

Page 17, line 27: There should be also PSCs lower down, at least to 80hPa.

Section 4.5: $H_2O$ is very sensitive to uncertainties in the parameterized satellite data and gap filling in the lowermost stratosphere, please discuss. This section is difficult to understand.

Page 21, line 3: Here the artificial heating/cooling from nudging can cause artifacts.

Page 22, line 9ff: Here the sensitivity study without temperature nudging between 10 and 100hPa should be discussed. This paragraph is too tentative now.

**3 Technical corrections**

Page 1, line 8: better "background" here.

Page 1, line 23: "photolysis of $O_2$".

Page 2, line 16: "reduction" instead of "loss".

Page 3, line 27: Is this the correct meaning of the acronym? There are several versions around in the literature, also it differs from the abstract.

Figures 1 and 2: Better use a logarithmic color scale instead of the scale with the arbitrary jump by one order of magnitude at $10\mu m^2/cm^3$.

---

## Author Comment (AC2) · 5 May 2020

We thank referee 1 for the quick report. Here are our replies to the comments:

• The introduction does a good job of explaining the context of your study and exactly what the knowledge gaps are.

Reply: Thank you very much for your positive feedback!

• However, what was missing is why those knowledge gaps are important to address. More specifically, why is it important to separate the chemical effect of

heterogeneous chemistry from the heating effect of the volcanic heating? I think a sentence or two would be useful here.

Reply: Volcanic aerosols in the stratosphere modify the atmosphere in two ways: they increase the surface area for heterogeneous reactions and modify the radiation budget by absorbing shortwave incoming sunlight and subsequently heat the stratosphere, resulting in a stronger transport of heat and ozone towards the poles. A reduction of degrees of freedom in a model simulation ("separation of effects") helps to understand the interplay of different processes and allows for estimating the respective quantitative contributions (transport versus chemical change in ozone) to the total effect in the ozone budget. Well, overall it is an academic question, but the separation helps to advance our understanding of the physical and chemical processes and their interplay. We will clarify this in the revised manuscript.

Moreover, this advanced knowledge is possibly relevant for geo-engineering by stratospheric sulfur injections, but we are hesitating to open a discussion about this here or in our manuscript, because it is clearly out of scope.

• Since you're using prescribed surface area density, it might be useful to talk about which processes/feedbacks you're missing that might affect your results.

Reply: With prescribed aerosol distribution we miss the primary feedback of the aerosol induced heating on the transport of the aerosol. But since we are interested more in the chemical effects here, this choice is advantageous, because the aerosol distribution is derived from observations. Yet, this approach might mask errors in the transport (incl. sedimentation) of aerosol of the model. We add this information to the revised manuscript (Discussion).

• It would be useful to talk about how the nudging may have affected your results. For example, on Page 7 you talk about transport of SWV, increases in vertical Interactive comment

motion, and polar stratospheric clouds. Are these things affected by the nudging? Are others?

Reply: SWV, strengthening of the vertical motion, and polar stratospheric clouds are influenced by nudging only indirectly. Nudging (to ERA-interim analysis data) is applied to the prognostic variables temperature, divergence and vorticity (-> horizontal wind field), and the logarithm of the surface pressure. We apply this nudging in the spectral space by omitting the nudging of wave-zero of the temperature, thus we do not correct temperature biases, implying that the absolute temperature can evolve. Moreover, the nudging is applied as low-normal mode insertion, i.e. down to the synoptic scale only, with comparably long relaxation times. This means, that the synoptic situation is that of ERA-Interim, whereas sub-synoptic variations can evolve freely. Such as for instance the influence of the volcanic cloud on the vertical velocity and the temperature profile. This can be clearly seen in the results. We will discuss the effects of our nudging procedure in the revised manuscript in more detail. We will also refer to our sensitivity simulation with respect to nudging as described by Löffler et al., 20161 (see their Supplement and the open review discussion).

• There are quite a few typos. I've pointed out some but not others. I'd recommend the authors spend some time proofreading.

Reply: Yes, we will proofread the manuscript again.

Specific Comments

• Page 2, line 12: Poleward misspelled
<sup>1Löffler, M., Brinkop, S., and Jöckel, P.: Impact of major volcanic eruptions on stratospheric water vapour, Atmos. Chem. Phys., 16, 6547–6562, https://doi.org/10.5194/acp-16-6547-2016, 2016.

Reply: Yes, corrected!

• Page 5, lines 2-5: These read like throwaway comments. What are the sociopolitical impacts?

Reply:

"Besides the scientific relevance, the obtained results are supposed to also have political and social impacts. This is especially important for the contribution to the WMO/UNEP (World Meteorological Organization/United Nations Environment Programme) ozone and IPCC (Intergovernmental Panel on Climate Change) climate assessments (WMO, 2019)."

Indeed. claiming a sociopolitical impact here is a bit exaggerated. We will remove this statement from the revised manuscript and reduce our comment to the relevance of a better process understanding for the WMO/UNEP and IPCC assessments.

• Page 5, line 5: Assess misspelled

Reply: Yes, corrected!

• Page 10, line 1: Is this really exactly linear? I think you need to demonstrate this more clearly.

Reply: At this point, we just wanted to clarify the superposition of both simulations, which indeed does not necessarily be linear. Thus, we will remove the word "linear".

---

## Short Comment (SC2) · 25 May 2020

Dear authors,

this short comment is mainly concerned with the simulated volcanic heating which you stated to be of 3.5-4 K between 50-60 hPa in VOL agreeing well with results from Labitzke and McCormick (1992), who observed a stratospheric heating at 50 hPa of 3-4.5 K in the tropics. Angell (1997a), who used radiosonde data, figured out a temperature increase of approximately 3-4 K between 30-50 hPa in late 1991. However, more recent studies (Revell et al, 2017; see Fig. 4 in Kuchar et al, 2017) showed that simulations using the CCMI aerosol data set overestimate the temperature response to the Mt Pinatubo eruption and novel

CMIP6 stratospheric aerosol data are in excellent agreement with MERRA and ERA-Interim reanalyses. Whether you want to compare your results with these studies, model simulation datasets are available via British Atmospheric Data Centre; see http://catalogue.ceda.ac.uk/uuid/1005d2c25d14483aa66a5f4a7f50fcf0 or at https://data.mendeley.com/datasets/khrhbw6wn5/1 (Kuchar and Revell, 2017).

Please consider these facts in the discussion of your results.

Best regards

Ales Kuchar

References

Kuchar, A., Ball, W. T., Rozanov, E. V., Stenke, A., Revell, L., Miksovsky, J., Pisoft, P., and Peter T.: On the aliasing of the solar cycle in the lower stratospheric tropical temperature, J. Geophys.Res.-Atmos., 122, 907–9093, doi:10.1002/2017JD026948, 2017.

Kuchar, A. and Revell, L.: SOCOLv3 model data for "On the aliasing of the solar cycle in the lower-stratospheric tropical temperature" and "Chemistry-climate model simulations of the Mt. Pinatubo eruption using CCMI and CMIP6 stratospheric aerosol data sets", https://doi.org/10.17632/khrhbw6wn5.1,2017.

Revell, L. E., Stenke, A., Luo, B., Kremser, S., Rozanov, E., Sukhodolov, T., and Peter, T.: Impacts of Mt Pinatubo volcanic aerosol on the tropical stratosphere in chemistry-climate model simulations using CCMI and CMIP6 stratospheric aerosol data, Atmos. Chem. Phys.,17, 13 139–13 150, https://doi.org/10.5194/acp-17-13139-2017, https://www.atmos-chem-phys.net/17/13139/2017/, 2017.

---

## Author Comment (AC3) · 24 Jun 2020

Dear authors, this short comment is mainly concerned with the simulated volcanic heating which you stated to be of 3.5-4 K between 50-60 hPa in VOL agreeing well with results from Labitzke and McCormick (1992), who observed a stratospheric heating at 50 hPa of 3-4.5 K in the tropics. Angell (1997a), who used radiosonde data, figured out a temperature increase of approximately 3-4 K between 30-50 hPa in late 1991. However, more recent studies (Revell et al, 2017; see Fig. 4 in Kuchar et al, 2017) showed that simulations using the CCMI aerosol data set overestimate the temperature response to the Mt Pinatubo eruption and novel CMIP6 stratospheric aerosol data are in excellent agreement with MERRA and ERA-Interim reanalyses. Whether you want to compare your results with these

studies, model simulation datasets are available via British Atmospheric Data Centre; see http://catalogue.ceda.ac.uk/uuid/1005d2c25d14483aa66a5f4a7f50fcf0 or at https://data.mendeley.com/datasets/khrhbw6wn5/1 (Kuchar and Revell, 2017).

Please consider these facts in the discussion of your results. Best regards Ales Kuchar

- *Thank you very much for your critical remark. Indeed, model simulations using the CCMI aerosol data might overestimate the volcanic heating. We will add this point to our discussion and refer to your study (Kuchar and Revell, 2017).*

---

## Author Comment (AC4) · 24 Jun 2020

We thank referee 2 for the quick report. Here are our replies to the comments:

- *General Comments: The paper tries to separate the effects of aerosol heating and aerosol chemistry on ozone. It is, however, an odd concept to nudge temperatures and winds to observations in the region where temperature and dynamics changes due to aerosol should be analysed (page 5 and 6). This must have consequences for the results. To get some feeling for the introduced artifact it would be good to perform a sensitivity simulation with nudging only up to about 100hPa to get the tropospheric wave forcing but the unperturbed effects of calculated radiative heating due to Pinatubo aerosol on stratospheric dynamics, QBO nudging*

[Figure]

*based on the Singapore data might be on for this case. The manuscript might be
published after revision.*

Reply: Thank you for your feedback. However, we disagree that we use an "odd"
concept to study the Mt. Pinatubo eruption with nudged simulations. This concept has
explicitly been selected as appropriate due to the following reasons:

Nudging (to ERA-interim analysis data) is applied to the prognostic variables temper-
ature, divergence and vorticity (-> horizontal wind field), and the logarithm of the sur-
face pressure. We apply this nudging in the spectral space by omitting the nudging of
wave-zero of the temperature, thus we do not correct temperature biases, implying that
the absolute temperature can evolve. Moreover, the nudging is applied as low-normal
mode insertion, i.e. down to the synoptic scale only, with comparably long relaxation
times. The nudging is applied such that the large(r than synoptic) scale patterns corre-
spond to those of ERA-Interim, but not the absolute temperature. This means that the
synoptic situation is that of ERA-Interim, whereas sub-synoptic variations can evolve
freely. Such as for instance the influence of the volcanic cloud on the vertical velocity
and the temperature profile.

The effect of nudging on the results was already discussed in detail in a previous study
on the Mt. Pinatubo eruption by Löffler et al. (2016, their sections 2.3 and 5, see also
their supplement). In that study the scientific focus was on the change of water vapour
due to the eruption. During the review process of the study the question came up, how
the nudging of temperatures might influence the results. This point is indeed important,
as the water vapour change due to the eruption strongly depends on the temperature
distribution at the cold point and hence on how the aerosol heating of the volcanic cloud
is represented in a nudged simulation. As part of that study, an additional set of sen-
sitivity simulation pairs (one simulation with (VOL) and one without volcanic eruption
(NOVOL)) with prescribed monthly average chemistry (to save computing time) was
performed to address the sensitivity of the model results to the nudging procedure. The

simulation pair we are interested in (called QF – quasi free running simulation) nudged only the (logarithm) of the surface pressure (and prescribed SST/SIC) to study the effect of omitting the nudging of temperature, divergence and vorticity on the results. The presented results are all differences between a simulation with volcano (VOL) and without (NOVOL) in the specific model configuration, i.e. for our case the QF pair. This previous sensitivity study does not exactly use the nudging height as proposed by the referee, but represents a simulation where the model is nearly free-running, but does not deviate too far from the actual synoptic situation due to the surface pressure nudging. The effect of the volcanic eruption on the temperature in different heights in the stratosphere can be seen in the Figure S2 below (taken from the supplement of Löffler et al., 2016). The temperature change (VOL-NOVOL) for QF is overestimated compared to the nudged simulations and, moreover, it appears noisy. However, the development of temperature is similar to the nudged simulation pairs (FC-full chemistry and RE-prescribed chemistry). For more details we refer to Section 2.3 and Section 5 (page 6557) of Löffler et al. (2016).

Using nudged simulations has several advantages over free-running simulations to study the impact on the chemistry: The temperature response is closer to observations, which is important, as ozone chemistry is temperature dependent. The results appear less noisy. Our (nudged) simulation pair (VOL and NOVOL) are similar with respect to the synoptic situation, so the effect of aerosol heating on subgrid-scale chemistry and transport of ozone can be contoured more clearly. This would be more difficult, if one allows the synoptic situation to evolve freely. Moreover, for such a concept a large set of ensemble simulations is necessary (see for instance Aquila et al., 2013).

We will add the paragraph (item 3 from above) to the description of our methodology (section 2.3) to the revised manuscript.

We agree, however, that our concept is not appropriate to study the effect of the volcanic eruption on the global dynamical system, as it has been studied by Graf et al. (1993) and this was not our intention, either. Although we thought that we have mentioned that in our manuscript, it was obviously not clear or detailed enough. We will also refer to our sensitivity simulation with respect to nudging as described by Löffler et al., 2016 (see their Supplement and the open review discussion).

> *2 Specific comments:Page 1, line 5: The use of CCMI slang hides the problem with nudged temperatures up to 10hPa.*

Problems with "specified dynamics"and/or "T42L90MA"? The used wording corresponds to the standard expressions used in modeling and is further explained in the main text. We already discussed the influence of temperature nudging further above (page 1).

> - *Page 1, line 10f: Separate between aerosol and ozone heating (see section 4.1), please reword for clarity.*

Reply: Thank you for this hint. We will clarify the difference between the volcanic heating by the aerosol and the secondary heating effect due to the ozone increase in the revised manuscript. We will add the following explanation to section 4.1:

"The strongest heating due to absorption of solar and terrestrial infrared radiation by volcanic aerosols and by the increase of ozone due to transport occurs in the middle stratosphere of thetropics (Figure 4b)."

> - *Page 2, line 8: Don't forget to mention the terrestrial infrared.*

Reply: We will mention the terrestrial infrared radiation in the introduction of the revised manuscript as suggested.

> - *In the introduction references to earlier studies with EMAC on Pinatubo are missing.*

Reply: We forgot to cite the study by Brühl et al. (2015), who presented an inter-comparison between observations and model simulations of the stratospheric sulfur cycle and its relation to radiative and dynamical processes. We will add the reference to the revised introduction.

- *Page 4, line 6: Is the assumed distribution monomodal or how many modes are included?*

Reply: Thank you for your comment. The assumed distribution is a single-mode log-normal aerosol size distribution (Revell et al., 2017). We will add this information to the revised manuscript.

- *Page 5, line 17: 2011 is perturbed by the medium size volcanic eruption of Nabro and therefore not background. Where are the data from? SAGE died in 2005.*

Reply: Thank you for your critical question. Indeed, you are right, the aerosol data of our NOVOL simulation originate from the CCMI dataset, more precisely from CALIPSO (2006-2012) and not from SAGE (which terminated operation in 2005) (Diallo et al. 2017; Revell et al., 2017). The definition of a "background" aerosol distribution refers to a period, when volcanic activity and related aerosols are less present. We selected the year 2011 because no strong eruption occurred in the decade before. Nevertheless, the medium size volcanic eruption of Nabro in 2011 is indeed included in the CCMI dataset. In comparison with other eruptions Nabro just emitted 1 Tg SO2 (Pinatubo: 20Tg) into the atmosphere up to heights of 14 km. The tropopause height at 15° N is at 17 km and thus above the volcanic plume. Therefore we assume that only a negligible amount of sulphate aerosols were emitted into the stratosphere in that year. We will correct this point in section 2.2 and discuss this in our revised manuscript.

- *Page 7, line 18: More details please, NIR, IR. Transport effect on ozone heating?*

Reply: Thank you for your comment. The assumed distribution is a single-mode log-normal aerosol size distribution (Revell et al., 2017). We will add this information to the revised manuscript.

- *Page 9, section 4.2: Why is 1993 not addressed? In this year were the largest effects on total ozone in midlatitudes in observations but also in Fig. 5. It would be also good to compare with observations here (e.g. TOMS).*

Reply: You are right. Overall, 1993 shows the strongest decrease of the ozone column after the eruption. So from the point of view of the observations, this might be the most interesting year to analyze. The separation of the effects, however, shows that the chemistry effect, resulting in strong ozone depletion, has the largest impact already in 1992. Nevertheless, most of our analysis is represented as a time series, thus the year 1993 is already considered in the presentation of the results. We will compare the simulated total ozone in the midlatitudes with the TOMS observations in the revised section 3. Thank you for the hint.

- *Page 10, line 1: This appears to be in contradiction to Fig. 4.*

Reply: Thank you for your comment. This might be a misunderstanding. The differences between the effects (chemistry and heating effect) are additive: the combined effect (VOL-NOVOL) is the sum of the heating effect (VOL-CVOL) and the chemical effect (CVOL-NOVOL). This can be shown for total ozone, but also for the temperature in Figure 4. Due to a remark from reviewer1 we eliminated the word "linear" in line 2 of Page 10 and hope that this clarifies this misunderstanding.

- *Page 10, line 31: You may mention PSCs here.*

Reply: Thank you very much for your feedback. We will refer here to the formation of PSC's and give a reference to section 4.4.1 in the revised manuscript.

- *Page 12: There are several ways to separate the catalytic ozone destruction cycles. You may discuss the meaning of HO2 + NO for aerosol perturbed lower stratospheric ozone.*

Reply: It is not really clear to us, what the referee suggests here. The reaction HO2 + NO —> OH + NO2 is a gas-phase reaction which affects the interconversion between HO2 and OH, in other words there ratio at all altitudes. Their ratio is controlled by the temperature dependent reaction rate and through the concentrations of O3 and NO. In the lower stratosphere the reaction HO2 + O is negligible, and HO2 + O3 is predominant. Since NO decreases in the winter 1991 in the lower stratosphere in the tropics and in spring 1992 in higher latitudes, more HO2 is available to react with O3 instead of with NO. This leads to an increase of the meaning of the HOx cycle for the lower stratosphere.

We have added this explanation into section 4.4 of the revised text.

- *Figures 10 and 11 might be merged, as well as Figs. 12 and 13. It should be also better to show a common pressure level in the lower stratosphere in tropics and extratropics (Figs. 9 to 13) instead of 30 and 20hPa. What is included in Ox? Please define, the Chapman cycle alone cannot explain the curves for the lower stratosphere.*

Reply: Indeed, it is a good idea to merge Figures 10 and 11, and Figures 12 and 13 into one figure, respectively. We will modify the manuscript accordingly. However, we are very much hesitating to show common pressure levels in Figures 9-13. The vertical levels were selected due to the results displayed in Figures 6-8, because at 20 and 70

hPa the strongest ozone decrease/increase could be found and therefore these levels appeared to be most interesting to analyze. Yet, we will motivate this in the revised manuscript.

Our study, includes all relevant Chapman equations in the Ox cycles such as the production of ozone via $O2 + hv$ —> $O + O$ and $O + O2$ —> $O3$, as well as the ozone depletion via $O3 + hv$ —> $O2 + O$ and $O + O3$ —> $2O2$. Note that the photolysis rates in our study are unaffected by the volcanic aerosol. Hence, the photolysis of ozone might be overestimated.

We do not understand the last point you addressed, that "the Chapman cycle alone cannot explain the curves for the lower stratosphere". Nowhere, we claimed that the Chapman cycle alone explains the curves for the lower stratosphere. In Figures 11 and 13 we find the ClOx, BrOx and HOx cycles being important in reducing ozone, when the NOx cycle is reduced by the volcanic aerosols.

- *Page 17, line 27: There should be also PSCs lower down, at least to 80hPa.*

Reply: You are right. The data in Figure 14 are the summed volume of PSCs down to 80 hPa. We will clarify this in the revised manuscript.

- *Section 4.5: H2O is very sensitive to uncertainties in the parameterized satellite data and gap filling in the lowermost stratosphere, please discuss. This section is difficult to understand.*

Reply: Thank you for your feedback. We agree that the uptake of water vapour by aerosols is sensitive to the aerosol surface area density and consequently affected by uncertainties in the satellite data and gap filling of the CCMI dataset. We will mention this uncertainty in the discussion of the revised manuscript.

- *Page 21, line 3: Here the artificial heating/cooling from nudging can cause artifacts.*

We apply nudging in the spectral space by omitting the nudging of wave-zero of the temperature, thus we do not correct temperature biases, implying that the absolute temperature can evolve. Moreover, the nudging is applied as low-normal mode insertion, i.e. down to the synoptic scale only, with comparably long relaxation times. The nudging is applied such that the large(r than synoptic) scale patterns correspond to those of ERA-Interim, but NOT the absolute temperature. This means, that the synoptic situation is that of ERA-Interim, whereas sub-synoptic variations can evolve freely, such as for instance the influence of the volcanic cloud on the vertical velocity and the temperature profile. We will extent the discussion and display the results from Löffler et al,(2016) on the comparison of a nudged and free-running simulation of the Mt. Pinatubo eruption.

- *Page 22, line 9ff: Here the sensitivity study without temperature nudging between 10 and 100hPa should be discussed. This paragraph is too tentative now.*

We will discuss and refer to our results from the quasi-free running simulation as presented and discussed by Löffler et al. (2016) in the revised manuscript. In that study the effect of nudging on the results of the Mt. Pinatubo eruption was already presented. We agree that this aspect is so important that it should be addressed in this study, too.

- *3 Technical corrections Page 1, line 8: better "background" here.*

Reply: Yes, you are right we will reword this.

- *Page 1, line 23: "photolysis of O2".*

Reply: Thank you, we added O2.

- *Page 2, line 16: "reduction" instead of "loss".*

Reply: Yes, corrected.

- *Page 3, line 27: Is this the correct meaning of the acronym? There are several versions around in the literature, also it differs from the abstract.*

Reply: We will use the following description for our model: Version 2.51 of the European Centre for Medium-Range Weather Forecasts-Hamburg (ECHAM)/Modular Earth Submodel System (MESSy) Atmospheric Chemistry (EMAC) model. We will correct it in the revised manuscript.

- *Figures 1 and 2: Better use a logarithmic color scale instead of the scale with the arbitrary jump by one order of magnitude at $10\mu m2/cm3$.*

We do not really understand you criticism. The shown figure of the aerosol surface density has a logarithmic scale except for the range < 1 because these values are showing the background aerosol and are not important at this point. We think, that our representation is an appropriate way to highlight the volcanic aerosol plume.

**References:**

Aquila, V., Oman, L. D., Stolarski, R., Douglass, A. R., and Newman, P. A.: The Response of Ozone and Nitrogen Dioxide to the Eruption 5 of Mt. Pinatubo at Southern and Northern Midlatitudes, Journal of the Atmospheric Sciences, 70, 894–900, https://doi.org/10.1175/JASD-12-0143.1, https://doi.org/10.1175/JAS-D-12-0143.

Brühl C, Lelieveld J, Tost H, Höpfner M, Glatthor N. Stratospheric sulfur and its implications for radiative forcing simulated by the chemistry climate model EMAC. J Geophys Res Atmos. 2015;120(5):2103‐2118. doi:10.1002/2014JD022430013.

Diallo, M., Ploeger, F., Konopka, P., Birner, T., Müller, R., Riese, M.,. . . Jegou, F. ( 2017). Significant contributions of volcanic aerosols to decadal changes in the stratospheric circulation. Geophysical Research Letters, 44, 10,780– 10,791. https://doi.org/10.1002/2017GL074662

Graf, H.-F., I. Kirchner, A. Robock, and I. Schult, Pinatubo eruption winter climate effects: Model versus observations, Clim. Dyn., 9, 81-93, 1993.

Löffler, M., Brinkop, S., and Jöckel, P.: Impact of major volcanic eruptions on stratospheric water vapour, Atmos. Chem. Phys., 16, 6547– 6562, https://doi.org/10.5194/acp-16-6547-2016, https://www.atmos-chem-phys.net/16/6547/2016/, 2016

Revell, L. E., Stenke, A., Luo, B., Kremser, S., Rozanov, E., Sukhodolov, T., and Peter, T.: Impacts of Mt Pinatubo volcanic aerosol on thetropical stratosphere in chemistry– climate model simulations using CCMI and CMIP6 stratospheric aerosol data, Atmos. Chem. Phys., 17, 13 139–13 150, https://doi.org/10.5194/acp-17-13139-2017, https://www.atmos-chem-phys.net/17/13139/2017/, 2017.
* * *
[Figure]

[Figure]

Figure S2: Temperature [K] differences (VOL-NOVOL) for the tropics (5°S-5°N), zonally averaged after the June 1991 Mount Pinatubo eruption for 20 hPa (upper panel), 30 hPa (middle panel) and 50 hPa (lower panel). The different simulation pairs are coloured as labelled in the upper panel.

**Fig. 1.**

---

## Author Response (AR2)

*Comments to the Author:*
*Dear authors,*

*please find enclosed an referee report on your revised version of the manuscript. There are still some issues that need to be fixed before the manuscript can be accepted for publication in ACP. Further, I would like to ask you to consider additionally the following comments/corrections:*

*P2, L10: He -> They*

      Reply: Yes, indeed. Corrected.

*P6, L25: remove parantheses around Löffler.*

      Reply: Done.

*P7, L6: Here you could also consider to cite one of my papers where EMAC has been evaluated for the Arctic polar stratosphere (Khosrawi et al., 2017 and Khosrawi et al. 2018, both ACP).*

      Reply: Thank you for the hint. The papers are cited in the evaluation section of the revised document

*P8, L1: "total" appears twice, thus one is obsolete*

      Reply: Corrected.

*Best regards, Farahnaz Khosrawi*

*referee#2*

*General:*

*The manuscript has improved a lot concerning clarifications and additional information.*
*It is however odd that some improvements were promised in the open discussion but are not included in the revised manuscript. This holds for the technical corrections, merging of figures, but also remarks on uncertainties in water vapor. The discussion of the impact of nudging on the results in the region of interest should be longer including sentences of the open discussion. The supplement of Löffler et al. (2016) points to the fact that calculated temperature differences are dampened by nudging when comparing the nudged and the free running simulations. Nudging introduces always additional heating and cooling terms, sometimes in an indirect way via dynamical coupling, even if the global average temperature at the level is not nudged. This view is shared with the expert for the ECHAM nudging code, Ingo Kirchner of Free University of Berlin. Nudging up to 100hPa, i.e. below the region where the aerosol heating occurs, together with weak QBO nudging, would be a cleaner solution and avoids the noisy and expensive free running ensembles.*

> Reply: Thank you for your critical remarks. Indeed, some technical corrections have unfortunately been overlooked and they are now considered in the revised manuscript.
> We finally refrained from the proposed merging of the figures (Figs. 10 and 11, as well as 12 and 13 of the last version), because the merged figures would become too unclear. Instead, we moved the corresponding panels into one Figure.
> We agree that the impact of nudging on the results is an important topic and we further improved this passage in the discussion section. Moreover, we now discuss the influence of the aerosol data set on the water vapour distribution more thoroughly.

*Specific:*

*Page 2, line 16: skip 'heterogeneous' here. The gas-phase reactions matter in this context (line 20 O.K.).*

> Reply: Yes. Corrected.

*Page 3, line 8: methane is not of interest here because its effects are small.*

Reply: You are right. Corrected.

*Page 4, line 11f: spell out all instrument names.*

Reply: Yes we spell out all instrument names in the revised manuscript.

*Page 6, line 6f: Nabro is a 'medium-size eruption' in the literature. Its SO$_2$-plume extends into the lower stratosphere up to about 19km as seen by the MIPAS instrument.*

Reply: Thank you, we have checked this again and cite now Griessbach et al. (2016).

*Page 6, line 26ff: This is misleading. Nudging perturbs the calculated temperature changes but not the aerosol heating rates. The temperature change in the free running pair is larger than in the nudged pairs (about 40\%, Figs. S1 and S2 of L\"offler et al., 2016). Please reword and extend this part.*

Reply: Yes you are right. We reworded this part and extended it.

*Page 8, line 8: better say 'Antarctic polar cap'. Are the corresponding data for the northern hemisphere not available? It would be better to show both hemispheres in Fig. 4.*

Reply: Yes, we reworded this sentence in the revised manuscript and we will show both hemispheres in Figure 4.

*Page 9, line 12: old text was better (both signs!).*

Reply: As we got it right we reworded the old text and added the signs of the temperature change, respectively.

"Thus, the main temperature change of this volcanic eruption arises mostly from radiative absorption by the volcanic aerosol superimposed by a cooling due to reduced ozone."

*Figure 12: point to the increase in ozone loss by halogens in 2 springs (late winters) in NH in text.*

Reply: Thank you, we will explain this increase in more detail:

"The reduction of NOx accelerates the ClOx and BrOx cycle in late winter of 1992 and 1993, where the ClOx and BrOx cycles reach their maxima in depleting O3 (Figure 11g,h). The increase of the ozone loss rate by the halogens is not counterbalanced by any other cycle, which explains the strongest decrease of ozone at 70 hPa in late winter of 1992 and 1993 (Figure 11 b)."

*Figure 15: which latitude range is that? Poleward of 60 degrees?*

Reply: Yes, we have added this information to the caption.

Additional reference:

Griessbach, Sabine & Hoffmann, Lars & Spang, Reinhold & Von Hobe, Marc & Müller, Rolf & 
[revised manuscript text omitted]